# Feedback regulation of heat shock factor 1 (Hsf1) activity by Hsp70-mediated trimer unzipping and dissociation from DNA

Szymon W Kmiecik[1] ⓘ, Laura Le Breton[1,†] & Matthias P Mayer[1,*] ⓘ

## Abstract

The heat shock response is a universal transcriptional response to proteotoxic stress orchestrated by heat shock transcription factor Hsf1 in all eukaryotic cells. Despite over 40 years of intense research, the mechanism of Hsf1 activity regulation remains poorly understood at the molecular level. In metazoa, Hsf1 trimerizes upon heat shock through a leucine-zipper domain and binds to DNA. How Hsf1 is dislodged from DNA and monomerized remained enigmatic. Here, using purified proteins, we demonstrate that unmodified trimeric Hsf1 is dissociated from DNA *in vitro* by Hsc70 and DnaJB1. Hsc70 binds to multiple sites in Hsf1 with different affinities. Hsf1 trimers are monomerized by successive cycles of entropic pulling, unzipping the triple leucine-zipper. Starting this unzipping at several protomers of the Hsf1 trimer results in faster monomerization. This process directly monitors the concentration of Hsc70 and DnaJB1. During heat shock adaptation, Hsc70 first binds to a high-affinity site in the transactivation domain, leading to partial attenuation of the response, and subsequently, at higher concentrations, Hsc70 removes Hsf1 from DNA to restore the resting state.

**Keywords** attenuation; DNA binding; entropic pulling; heat shock response; Hsp70

**Subject Category** Translation & Protein Quality

**The EMBO Journal (2020) 39: e104096**

## Introduction

The heat shock response (HSR) is an ancient transcriptional program, evolved in all organisms to cope with a wide variety of environmental, physiological, and developmental stressful conditions that induce an imbalance of protein homeostasis. This transcriptional program up-regulates hundreds and down-regulates thousands of genes in metazoa (Vihervaara *et al*, 2018). As the HSR was viewed as the paradigm for a homeostatic response, the mechanism of its activation and attenuation was intensively studied in the last four decades but still remains poorly understood at a molecular level (Morimoto, 2011; Joutsen & Sistonen, 2018).

The central role in regulating the HSR and restoring protein homeostasis in eukaryotic cells is fulfilled by the heat shock transcription factor 1 (Hsf1). Through this function, Hsf1 is in metazoa at center stage of many physiological and pathophysiological processes like post-embryonic development and aging, cancer, and neurodegeneration (Xiao *et al*, 1999; Dai *et al*, 2007; Steele *et al*, 2008; Mendillo *et al*, 2012; Kim *et al*, 2016). Metazoan Hsf1 consists of a DNA-binding domain (DBD), leucine-zipper-like heptad repeat regions A and B (HR-A/B) functioning as trimerization domain, a regulatory domain (RD), a third heptad repeat region (HR-C), and a transactivation domain (TAD) (Fig 1A; Rabindran *et al*, 1993). Recent structural studies in our laboratory revealed that RD and TAD are mostly unstructured (Hentze *et al*, 2016).

In non-stressed mammalian cells, Hsf1 is in monomer–dimer equilibrium in the nucleus and the cytosol (Vujanac *et al*, 2005; Hentze *et al*, 2016). During stress conditions, Hsf1 accumulates in the nucleus and forms oligomers that gain increased affinity for binding to so-called heat shock elements (HSE: nGAAn) arranged in inverted repeats of three and more units (Sarge *et al*, 1993; Westwood & Wu, 1993; Zuo *et al*, 1994). Hsf1 HR-C region serves as an intramolecular Hsf1 oligomerization repressor that operates as a temperature sensor, the unfolding of which is proportional to the severity and the length of the heat stress (Rabindran *et al*, 1993; Hentze *et al*, 2016). The setpoint of this temperature rheostat is dependent on the concentration of Hsf1, allowing nuclear transport and Hsf1 expression to alter the fraction of trimerized Hsf1 at any given temperature (Jin *et al*, 2015; Hentze *et al*, 2016; Paul *et al*, 2018; Tsukamoto *et al*, 2019).

Based on accumulated knowledge, several Hsf1 activation/attenuation mechanisms have been proposed (Anckar & Sistonen, 2011). The most prominent Hsf1 activity regulation model, the chaperone titration model, assumes that Hsf1 is sequestered and inactivated by molecular chaperones; Hsf1 activation follows the recruitment of the chaperones to stress-denatured proteins. Since Hsp70 was found to co-precipitate with Hsf1, Hsp70 was implicated in repressing Hsf1

1   Center for Molecular Biology of Heidelberg University (ZMBH), DKFZ-ZMBH-Alliance, Heidelberg, Germany
    *Corresponding author. Tel: +49 6221 546829; Fax: +49 6221 545894; E-mail: m.mayer@zmbh.uni-heidelberg.de
    †Present address: Bioconjugation Unit, Novasep Manufacturing Solutions, Le Mans, France

in the resting state or during attenuation (Abravaya *et al*, 1992). Other evidence suggested that Hsp70 is insufficient in metazoa for Hsf1 repression, contesting this model (Rabindran *et al*, 1994). Since loss of Hsp90 functionality activates the HSR, it was also suggested that Hsp90 chaperones sequester Hsf1 and keep it inactive (Zou *et al*, 1998). However, to demonstrate Hsp90 and Hsf1 interaction, crosslinking is required (Guo *et al*, 2001), unless an ATPase-deficient Hsp90 variant is used, indicating that Hsf1 interacts with Hsp90 only in the ATP-bound closed conformation (Prince *et al*,

2015; Kijima *et al*, 2018). In addition, *in vitro* and in the absence of co-chaperones, Hsp90 favors Hsf1 trimerization and DNA binding (Hentze *et al*, 2016).

Although genetic evidence suggested an involvement of Hsp90 in HSR regulation also in yeast (Duina *et al*, 1998), more recent *ex vivo* data suggest that Hsp70 is associated with Hsf1 under non-stress conditions and this interaction is disrupted upon heat shock (Zheng *et al*, 2016; Krakowiak *et al*, 2018). Whether such a model can be adopted for mammalian cells is not clear, since Hsf1 is constitutively

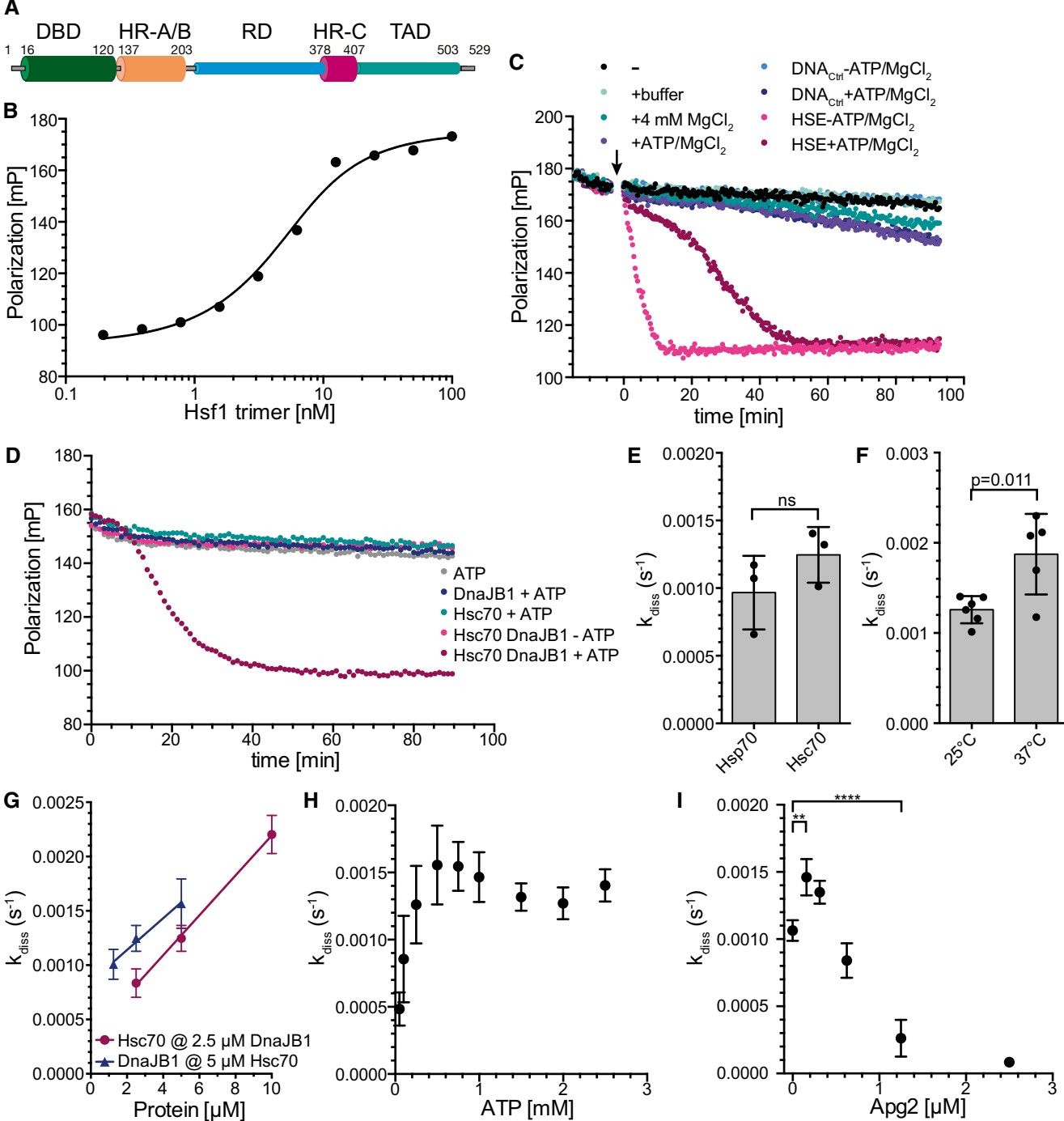

Figure 1.

◄

> **Figure 1. Hsf1 binds with high affinity but highly dynamic to heat shock elements (HSEs) and is dissociated from DNA by Hsc70 and DnaJB1.**
>
> A   Domain structure of mammalian Hsf1. Intrinsically disordered regions are indicated with cylinders of smaller diameter. DBD, DNA-binding domain; HR-A/B, heptad repeat regions A and B; RD, regulatory domain; HR-C, heptad repeat region C; TAD, transactivation domain.
>
> B   Equilibrium titration of Hsf1 binding to Alexa Fluor® 488-labeled HSE-DNA. Fluorescence polarization plotted versus the Hsf1 trimer concentration. $K_D = 4.7 \pm 1.9$ nM ($n = 4$).
>
> C   Hsf1 rapidly switches between different HSE-containing double-stranded DNA oligonucleotides. Trimeric Hsf1 was pre-incubated with Alexa Fluor® 488-labeled HSE-DNA in $MgCl_2$-free buffer. At timepoint 0 (arrow) buffer, 4 mM $MgCl_2$, 2 mM ATP+4 mM $MgCl_2$, control DNA ($DNA_{Ctrl}$ at 10-fold molar excess over HSE-DNA), and HSE-containing DNA were added as indicated and fluorescence polarization was followed over time. Shown is one of three identical experiments.
>
> D   Trimeric Hsf1 was bound to HSE-DNA and the indicated components added at timepoint 0. A representative experiment is shown.
>
> E   Hsp70 dissociates Hsf1 from DNA with similar rates as Hsc70. Rates were determined as shown in Fig EV1E. Shown are mean $\pm$ SD ($n = 3$). Significance was tested with Student's *t*-test.
>
> F–I   The rate of Hsc70/DnaJB1-mediated dissociation of Hsf1 from HSE-DNA depends on temperature (for 25°C $n = 6$, for 37°C $n = 5$, *t*-test, $P = 0.011$) (F), concentration of Hsc70 and DnaJB1 ($n = 3$) (G), concentration of ATP ($n = 3$) (H), and concentration of the nucleotide exchange factor Apg2 (HSPA4) ($n = 3$, ANOVA, **, $P < 0.01$; ****, $P < 0.0001$) (I). Shown are mean $\pm$ SD.
>
> Source data are available online for this figure.

trimeric in yeast and does not rely on a monomer–trimer transition for activation (Sorger *et al*, 1987). Moreover, the overall degree of sequence identity between yeast and human Hsf1 is just 17% and the proposed binding sites of Hsp70 in yeast Hsf1 are not conserved in human Hsf1.

Several different models have been proposed for HSR attenuation, Hsf1 dissociation from DNA, and recycling of Hsf1. Hsp70 and its co-chaperone DnaJB1 have been suggested to bind to Hsf1 within the TAD (aa 425–439), thus attenuating Hsf1 activity by repressing the recruitment of the transcriptional machinery (Shi *et al*, 1998). Hsf1 acetylation in the DBD was proposed to be required for removing Hsf1 from DNA (Westerheide *et al*, 2009). Thereby, the SIRT1 deacetylase plays a main role in delaying the attenuation of the HSR. The interaction between DNA and Hsf1 DBD relies on electrostatic contacts (Neudegger *et al*, 2016) and replacement of Lys80 and/or Lys118 in Hsf1 to glutamate significantly reduced DNA binding (Westerheide *et al*, 2009). Since the inhibition of proteasomal function also delayed HSR attenuation, it was suggested that, after acetylation-dependent dissociation, Hsf1 trimers are directed to proteasomal degradation (Raychaudhuri *et al*, 2014). However, inhibition of the proteasome also leads to accumulation of misfolded proteins in the cytosol, eliciting the HSR. For *Drosophila* Hsf1, it was shown that trimers disassemble spontaneously to monomers at low concentrations (Zhong *et al*, 1998). However, such a spontaneous dissociation was not observed for human Hsf1 (Hentze *et al*, 2016).

In this work, we demonstrate using purified components that both Hsc70 and Hsp70 in cooperation with DnaJB1 dissociate trimeric Hsf1 from DNA in the absence of Hsf1 acetylation and that during dissociation Hsf1 is monomerized. We identify several binding sites for Hsc70 within Hsf1, one of which in the transactivation domain involved in initial attenuation, a second close to the trimerization domain essential for Hsc70-mediated monomerization. We provide evidence that Hsc70-mediated monomerization of Hsf1 trimers occurs through stepwise unzipping of the triple leucine-zipper of the Hsf1-trimer by entropic pulling. To our knowledge, this is the first rigorous test of the entropic pulling hypothesis for the mode of action of Hsp70s. Mutational alteration of the Hsc70 binding sites potentiates expression of a heat shock reporter in $HSF1^{-/-}$ mouse embryonic fibroblasts, demonstrating the importance of our findings for the regulation of the Hsf1-mediated stress response. Based on these and published data, we propose a comprehensive mechanistic model for a dynamic regulation of Hsf1 activity that

closely monitors availability of cellular Hsc70 and Hsp70, explaining a large body of observations published over the last 40 years.

## Results

### Hsf1 can migrate rapidly between different HSE-containing DNAs

To investigate DNA binding of purified human trimeric Hsf1, we used the previously established fluorescence polarization assay with Alexa Fluor® 488-labeled double-stranded DNA oligonucleotides containing 3 inverted HSEs (Hentze *et al*, 2016). Fluorophores excited by polarized light emit light with the same polarization plane. If the fluorophore rotates between excitation and emission, the plane of polarization rotates with the dye. Therefore, fluorescence polarization monitors the rotational diffusion of a fluorescent molecule, attached to DNA in our case. Rates of rotational diffusion are high, and therefore, fluorescence polarization is low for free DNA. Binding of Hsf1 to the DNA decreases the rate of rotational diffusion and consequently increases fluorescence polarization. We first titrated Hsf1 and established the dissociation equilibrium constant $K_D$ to $\leq 5$ nM (Fig 1B) consistent with previous results (Hentze *et al*, 2016). The interaction of Hsf1 with HSE-containing DNA was very stable, and little decrease in fluorescence polarization was observed over 90 min (Fig 1C) when only buffer or excess unlabeled control DNA without HSEs was added. Addition of $MgCl_2$ (4 mM) without or with ATP or excess of control DNA led to a slow decrease in polarization, presumably due to slow dissociation and aggregation of Hsf1 during the long incubation time. In contrast, if excess of unlabeled HSE-DNA was added in the absence of $MgCl_2$ polarization decreased rapidly, reaching the polarization value of unbound labeled DNA within 10 min, indicating that Hsf1 can switch from one HSE-containing dsDNA segment to another at high rates. The decrease was significantly slower when the unlabeled HSE-DNA was added in the presence of $MgCl_2$, suggesting the $Mg^{2+}$ ions bound to the DNA backbone phosphate reduce the association rate of Hsf1 DBD to the DNA. From these experiments, we conclude that individual DBDs of the Hsf1 trimer dissociate from the HSE-DNA and re-associate rapidly in the absence of $Mg^{2+}$ ions and somewhat more slowly in the presence of $Mg^{2+}$ ions. For further experiments, we always pre-incubated trimeric Hsf1 with labeled HSE-containing dsDNA in the absence of $MgCl_2$ to

generate stably bound Hsf1-DNA complexes and then added different combinations of chaperones in the absence or presence of $Mg^{2+}$·ATP.

### Hsc70 but not Hsp90 can dissociate Hsf1 from DNA

A recent publication proposed an inhibitory role of Hsp90 during the attenuation phase of the HSR (Kijima *et al*, 2018). However, in our *in vitro* polarization DNA-binding assay neither human Hsp90α wild type nor its ATPase-deficient variant Hsp90α-E47A, which could be copurified much more efficiently with Hsf1 from transfected cells, had any influence on the change in polarization as compared to the controls (Fig EV1A), indicating that its effect during attenuation phase of the HSR was not achieved through dissociation of Hsf1 from DNA. This result is consistent with earlier findings that Hsp90 promotes Hsf1 trimerization and DNA binding (Hentze *et al*, 2016).

In contrast, human Hsc70 in the presence of ATP and its J-domain co-chaperone DnaJB1/Hdj1/Hsp40, which targets Hsc70 to client proteins by stimulating Hsc70's ATPase activity, efficiently dissociated Hsf1 from DNA (Fig 1D). This effect was not observed when any of the three components, Hsc70, DnaJB1, or ATP, was missing, or when Hsc70 wild type was replaced by its ATPase-deficient variant Hsc70-K71M, or its polypeptide binding defective variant Hsc70-V438F, or when DnaJB1 wild type was replaced by a variant (DnaJB1-H32Q,D34N/DnaJB1-QPN) that is not able to stimulate Hsc70's ATPase activity (Fig EV1B). Hsc70/DnaJB1-mediated dissociation of Hsf1 from DNA could also be demonstrated by an electrophoretic mobility shift assay (Fig EV1C and D).

When analyzing the shape of the dissociation curve, we observed a short 5- to 10-min delay, during which the dissociation did not follow an exponential function, before the actual exponential dissociation phase started. The data were therefore fitted by a composite function, and the rate only represents the exponential phase of the dissociation reaction (Fig EV1E). The dissociation rate was not significantly different whether we used Hsf1 purified from *E. coli* as trimer and not heat shocked or as monomer and subsequently heat shocked at 42°C for 10 min (Fig EV1F). Also, the heat-inducible Hsp70 (HSPA1A/B) dissociated Hsf1 from DNA with similar rates as the constitutive Hsc70 (HSPA8) (Fig 1E), and therefore, we have used Hsc70 for the remaining *in vitro* experiments, but believe that the result will also be valid for the heat-inducible Hsp70. The reaction was, as expected, temperature-dependent, and increasing the temperature from 25 to 37°C increased the dissociation rate significantly (Fig 1F). The kinetics of Hsc70-mediated Hsf1 dissociation from DNA were on the same time scale as the kinetics with which Hsf1-mediated transcription activation and DNA binding of Hsf1 decreased in HeLa cells during recovery after a heat shock (Abravaya *et al*, 1991).

The dissociation reaction rate was strongly dependent on the concentration of Hsc70 and DnaJB1, increasing almost threefold between 2.5 and 10 μM Hsc70 at 2.5 μM DnaJB1 and 1.5-fold between 1.25 to 5 μM DnaJB1 at 5 μM Hsc70 (Fig 1G).

The dissociation reaction was also strongly dependent on the ATP concentration between 0.05 and 0.5 mM, but not between 0.5 and 2.5 mM (Fig 1H). In the presence of physiological concentrations of ATP, the life time of an Hsc70-client protein complex is limited by nucleotide exchange that is accelerated by nucleotide

exchange factors (Mayer, 2013). We therefore added the nucleotide exchange factor Apg2 to the dissociation reaction. At very low concentrations, Apg2 accelerated the dissociation reaction, but at higher concentrations it strongly inhibited the reaction and prevented Hsc70-mediated dissociation of Hsf1 from DNA (Fig 1I). This is similar as Apg2 action in Hsc70-mediated protein disaggregation, where also low concentrations of Apg2 stimulate and high concentrations inhibit the reaction (Rampelt *et al*, 2012). Taken together, Hsc70 and Hsp70 dissociate Hsf1 from its binding sites in promoter DNA in the presence of DnaJB1 and physiological concentrations of ATP in a strongly concentration-dependent manner. This reaction is independent of Hsf1 acetylation in the DBD.

### Hsc70 dissociates Hsf1 from DNA by monomerization of Hsf1 trimers

For the Hsc70-mediated dissociation of Hsf1 from DNA, different modes of actions are imaginable. In analogy to Hsp70 action on p53 (Boysen *et al*, 2019; Dahiya *et al*, 2019), Hsc70 could directly interact with the DBD of Hsf1 to competitively or allosterically remove the DBD from the DNA or prevent rebinding of transiently dissociated DBD. Alternatively, Hsc70 could monomerize Hsf1, which would lead to dissociation from DNA because individual DBDs have only a very low affinity for the HSEs and the high affinity of Hsf1 for heat shock promoters is an avidity effect of three DBDs binding simultaneously. To test the second hypothesis, we followed the dissociation reaction by fluorescence polarization and incubated the same reaction mix in parallel for different time intervals, the reaction was stopped by adding blue-native sample buffer, and the samples were kept on ice before analyzing the oligomeric state of Hsf1 by blue-native polyacrylamide gel electrophoresis (BN-PAGE) and immunoblotting with Hsf1 specific antisera (Fig 2A and B). In the beginning of the reaction, only trimeric and higher order oligomeric Hsf1 was present. In the course of the dissociation reaction, the trimer band disappeared and some Hsf1 monomer became visible. Most of Hsf1 exhibited an electrophoretic mobility that is in between the trimeric and the monomeric state, possibly bound to Hsc70 or DnaJB1 or both. If any of the components were missing, Hsf1 remained oligomeric. Quantification of the Hsf1 oligomer band and the rest of the Hsf1 species revealed that the oligomer band disappeared within experimental error with the same rate as Hsf1 dissociated from DNA in the fluorescence polarization assay, strongly arguing that Hsc70 dissociates Hsf1 from DNA by monomerization. Hsc70 also monomerized trimeric Hsf1 with a similar rate in the absence of DNA (Fig 2C).

### Hsc70 binds to several sites in monomeric and trimeric Hsf1 and destabilizes the trimerization domain

To elucidate how Hsc70 dissociates Hsf1 trimers, we first wanted to identify the binding site of Hsc70 within Hsf1 using hydrogen exchange mass spectrometry (HX-MS), which is suitable to detect solvent accessibility of amide hydrogens of the peptide backbone and thus conformational changes in proteins and protein–protein interactions (Engen & Wales, 2015), as described in detail previously (Hentze & Mayer, 2013; Hentze *et al*, 2016). Briefly, we preincubated Hsf1 in the absence or presence of Hsc70 or DnaJB1 or both for 0 or 30 min at 25°C, diluted the sample 1:10 in $D_2O$

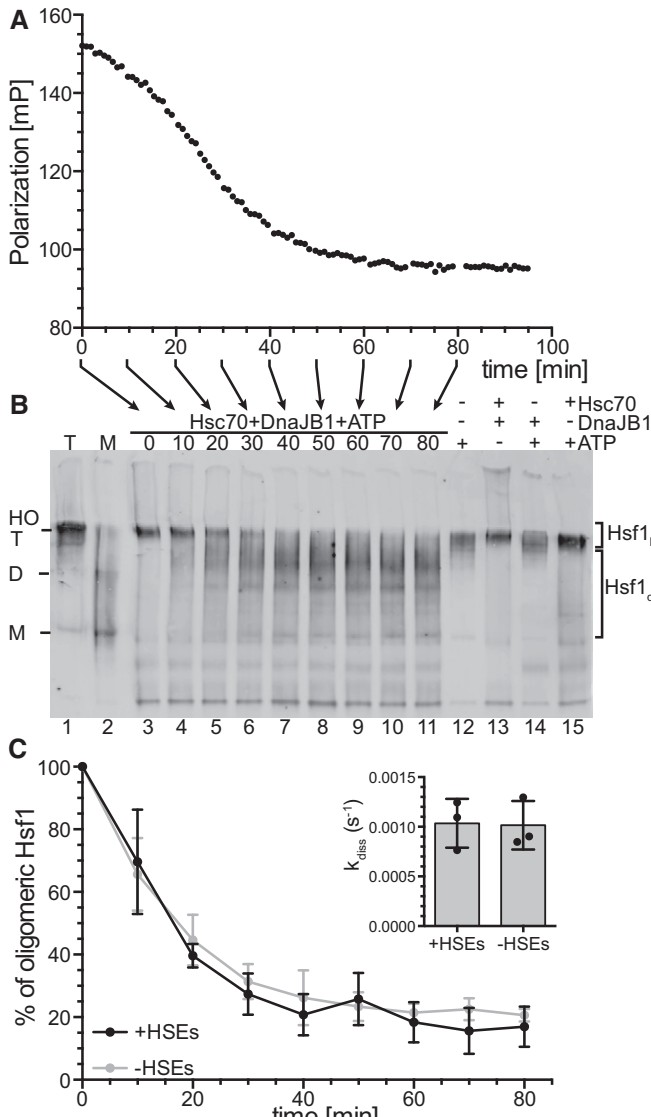

**Figure 2. Hsc70 and DnaJB1 dissociate Hsf1 from HSE-DNA by monomerization of the Hsf1 trimer.**

A, B  Standard Hsc70/DnaJB1-mediated dissociation reaction of Hsf1 from Alexa Fluor® 488-labeled HSE-DNA similar to Fig 1D monitored by fluorescence polarization (A). The same reaction mixture was incubated for different time intervals as indicated by the arrows below the *x*-axis, and the reaction stopped by addition of blue-native loading buffer, and stored on ice until separation by blue-native PAGE, blotted onto a PVDF membrane, and detected with an Hsf1-specific antiserum (B). Lanes 1, purified Hsf1 trimer (T); 2, purified Hsf1 monomer (M); 3–11, samples from the Hsc70/DnaJB1-mediated Hsf1 dissociation reaction (0–80 min); and 12–15, dissociation reaction incubated for 80 min missing individual components as indicated. HO, higher order oligomers; T, trimer; D, dimer; M, monomer.

C  Quantification of the amounts of Hsf1 species of the blot shown in (B) and five similar blots. For each lane, the intensities of the two areas indicated by the brackets to the right were integrated; upper bracket, DNA-bound timers and higher order oligomers (Hsf1_b); lower bracket, monomers and Hsf1-species possibly bound by Hsc70 or DnaJB1 or both dissociated from DNA (Hsf1_d). Shown are means ± SD (*n* = 3). Inset, deoligomerization rate as determined by fitting an exponential decay function to the data. Shown are means ± SD (*n* = 3).

Source data are available online for this figure.

containing buffer, and incubated for 30 or 300 s at 25°C. Subsequently, the exchange reaction was quenched and the samples were analyzed by LC-MS, including online digestion of the proteins by immobilized pepsin to localize the incorporated deuterons to specific segments of the protein.

Plotting the deuteron incorporation into Hsf1 in the presence of Hsc70 minus the deuteron incorporation in the absence of this chaperone revealed 5 regions of significant protection that were observed for trimeric as well as for monomeric Hsf1, suggesting 5 potential binding sites (Fig 3A left bar graph, Appendix Fig S1A and B). Surprisingly, we did not detect any protection close to amino acids 395–439 previously proposed to harbor the Hsc70 binding site in Hsf1 (Shi *et al*, 1998). All segments protected by Hsc70 were also protected by DnaJB1 (Fig 3A right bar graph, Appendix Fig S1C and D) consistent with the generally accepted mechanism of Hsp70 systems that J-domain proteins bind to the client protein first and target Hsp70 to its clients (Mayer, 2013). Of note, protection in HX-MS experiments does not necessarily mean direct binding. Protection could also be caused allosterically by changing the conformation of the client protein. Also, we may miss actual binding sites, if the region is not covered by the peptic peptides or if pepsin cleaves within the bound segment leading to a fast back-exchanging amino group. In addition, the protected segments were larger than the segment bound by the substrate binding domain of an Hsp70, introducing an uncertainty where exactly Hsc70 was bound within the segment. We therefore used an Hsp70 binding site prediction algorithm, originally derived from peptide library scanning data for the *E. coli* Hsp70 homolog DnaK (Rüdiger *et al*, 1997) to localize the potential Hsc70 binding site within the protected segment. Based on this algorithm, two of the segments (200–213 and 442–474) protected from hydrogen exchange by Hsc70 (indicated as thick horizontal black lines in Fig 4A underneath the prediction value curve) covered sequences that fitted properties of strong Hsp70 binding sites (Fig 4A, values below −5, dashed line). Close inspection of the sequence of these two segments revealed that both contained two potential Hsc70 binding sites.

To confirm Hsc70 binding to these sequences, we used peptides encompassing the respective sequences labeled with fluorescein and titrated Hsc70 concentration measuring fluorescence polarization. For all four potential binding sites in the two HX protected regions 202–213 and 442–474, we could determine a $K_D$ between 5 and 30 μM (Fig 4B). To peptides representing the most hydrophobic part of the other protected segments (N74-Q86-C, P223-H235-C, A343-R352-C), Hsc70 had a lower or no measurable affinity. The highest affinity Hsc70 binding site was in the TAD, residues 461–471 (Fig 4B).

When Hsc70, DnaJB1, and ATP were added to trimeric Hsf1 and pre-incubated for 30 min before dilution into $D_2O$ buffer and incubation for 300 s, we observed significant deprotection in three segments encompassing the trimerization domain (Fig 3B, Appendix Fig S2A), consistent with Hsc70-mediated monomerization. Close inspection of the respective spectra revealed a bimodal distribution of the isotopic peaks, indicative for the coexistence of two subpopulations with different exchange properties. Fitting an equation for two Gaussian distributions to the isotope peak intensities allowed to extract the fraction of high and low exchanging subpopulations (Fig 3C, Appendix Fig S2B and C). In the absence of chaperones, in the presence of ATP, or in the presence of Hsc70 and DnaJB1 but without ATP or with ATP but without pre-incubation only a small fraction of the peptides belonged to the high

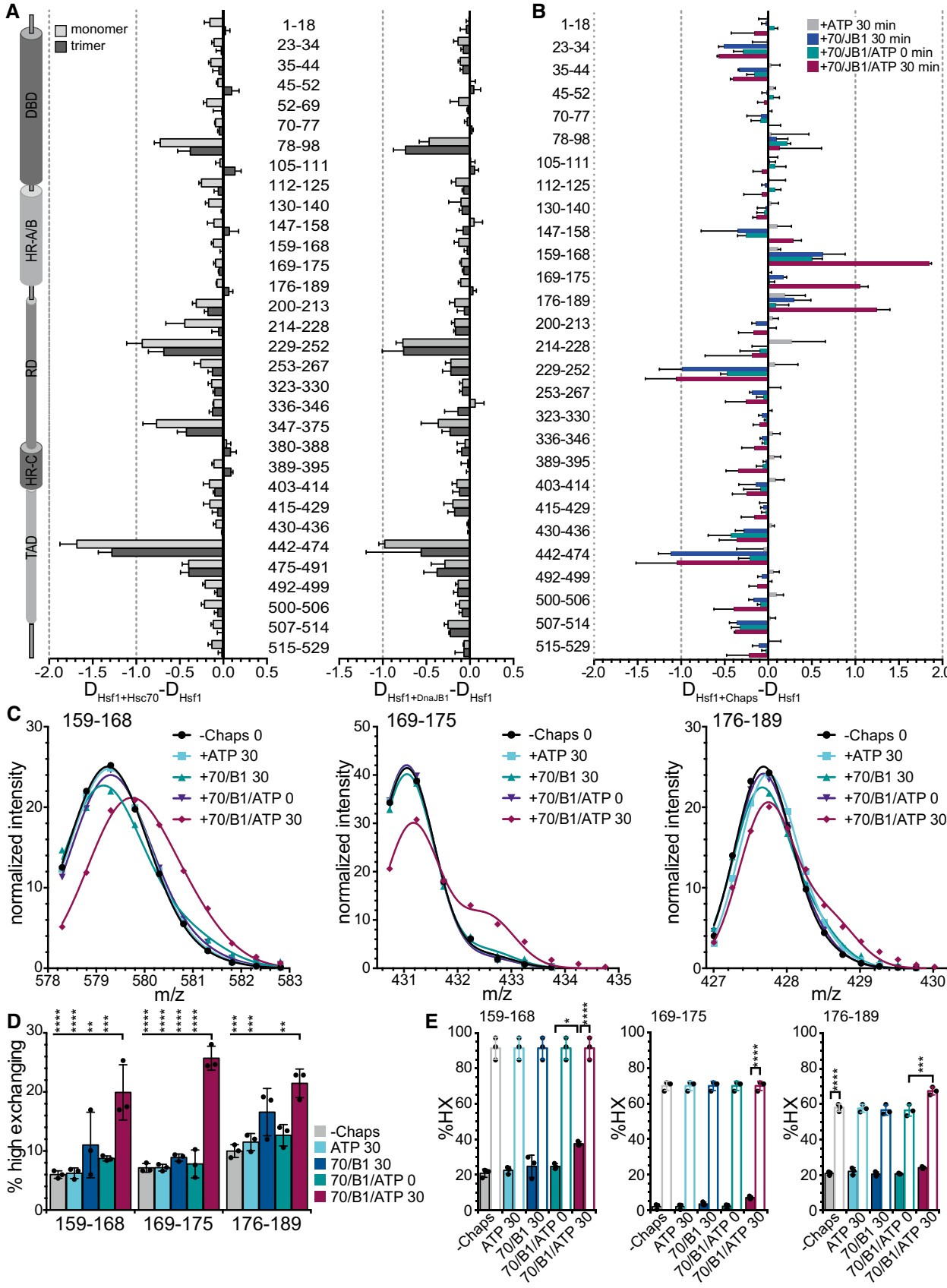

**Figure 3.**

**Figure 3.  Hsc70 and DnaJB1 interaction with Hsf1 trimers lead to deprotection of segments within the trimerization domain.**

A  Difference plot of deuteron incorporation into Hsf1 monomers (light gray) or trimers (dark gray) in the presence of Hsc70 (left) or DnaJB1 (right) minus deuteron incorporation into Hsf1 in the absence of chaperones after 30 s incubation in $D_2O$ buffer. Shown are mean $\pm$ SD ($n$ = 3) for peptic peptides as indicated between the panels. Left, Hsf1 domain representation. (see also Appendix Fig S1).

B  Difference plot of deuteron incorporation into Hsf1 trimers incubated for 300 s in $D_2O$ buffer after pre-incubation in the presence of ATP, Hsc70 (70), and DnaJB1 (B1) as indicated minus deuteron incorporation into Hsf1 in the absences of additives. For relative deuteron incorporation, see Appendix Fig S2A. Shown are means $\pm$ SD ($n$ = 3).

C  Fit of two Gaussian distributions to the peak intensities for peptic peptides 159–168 (left), 169–175 (middle), and 176–189 (right). Representative plots of three independent experiments. Original spectra and individual Gaussian distributions are shown in Appendix Fig S2B and C.

D  Fraction of high exchanging subpopulation for peptides 159–168, 169–175, and 176–189 under the different indicated pre-incubation conditions: -chaperones, +Hsc70 (70), +DnaJB1 (B1), and +ATP incubated for 0 or 30 min at 25°C. Shown are means $\pm$ SD ($n$=3). Statistical analysis: ANOVA with Sidak's multiple comparison of different conditions for each of the peptides; **$P$ < 0.01; ***$P$ < 0.001, ****$P$ < 0.0001.

E  Relative deuteron incorporation of the low (solid bars) and high (open bars) exchanging subpopulations. Shown are means $\pm$ SD ($n$ = 3). ANOVA, Sidak's multiple comparison test; *$P$ < 0.05; ***$P$ < 0.001; ****$P$ < 0.0001.

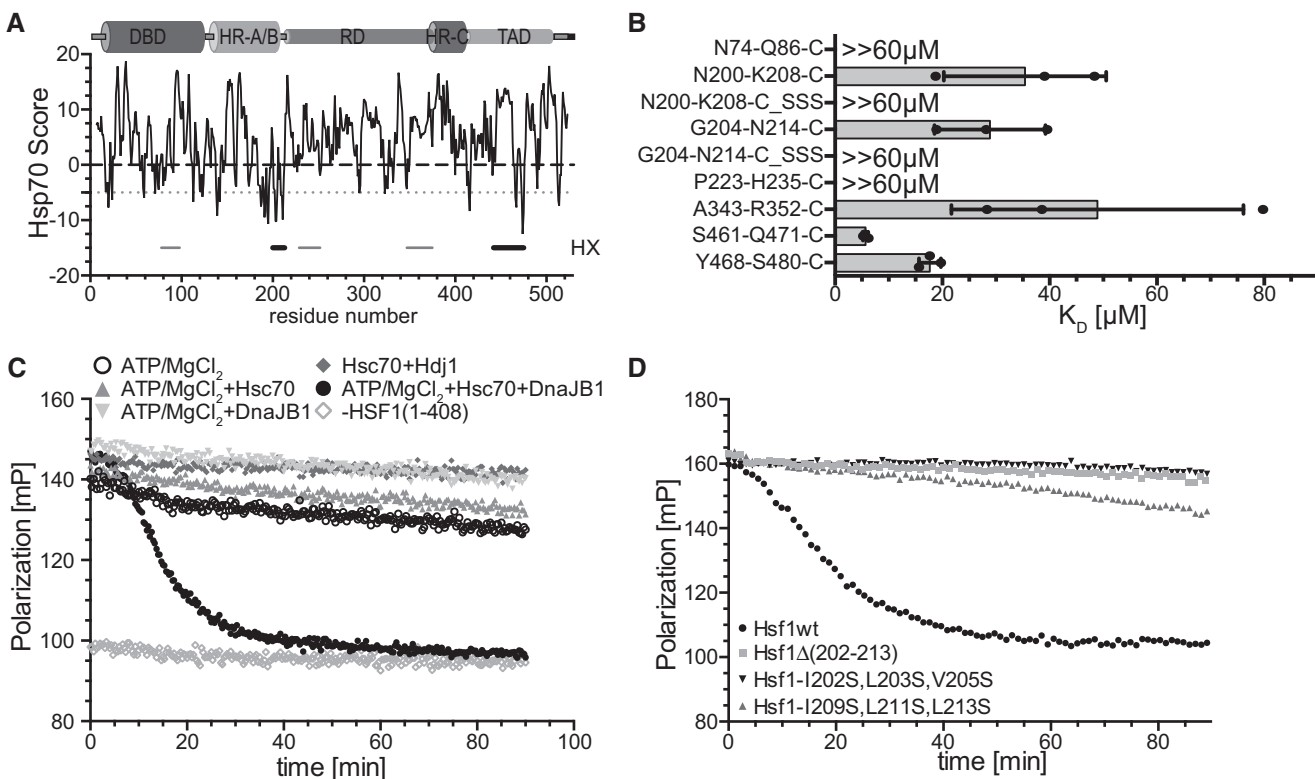

**Figure 4.  An Hsc70 binding site close to the trimerization domain of Hsf1 is necessary for Hsc70-mediated dissociation of Hsf1 from DNA.**

A  Hsp70 binding score as predicted by a published algorithm (Rüdiger *et al*, 1997). The score value is attributed to the center of the sequence window of 13 residues. Values below −5 are considered to be good Hsp70 binding sites. Small horizontal gray and black lines underneath the prediction curve labeled with HX to the right represent the segments protected from hydrogen exchange.

B  $K_D$ values for the indicated peptides as determined by fluorescence polarization. Peptides N74-Q86-C (NMYGFRKVVHIEQC), P223-H235-C (PKYSRQFSLEHVHC), and A343-R352-C (ASVTALTDARC) represent the most hydrophobic part of HX protected segments 78–98, 229–252, and 347–375, respectively; N200-K208-C (NRILGVKRKC), N200-K208-C_SSS (NRSSGSKRKC), G204-N214-C (GVKRKIPLMLNC), and G204-N214-C_SSS (GVKRKSPSMSNC) represent the two potential Hsc70 binding sites in segment 202–213 in original sequence or with hydrophobic residues replaced by serine; and S461-Q471-C (SGKQLVHYTAQC) and Y468-S480-C (YTAQPLFLLDPGSC) represent the two potential Hsc70 binding sites in segment 442–474. Shown are means $\pm$ SD ($n$ = 3).

C  Hsf1(1–408) with a deleted TAD is dissociated from HSE-DNA by Hsc70, DnaJB1, and ATP with similar rates as Hsf1wt. Fluorescence polarization assays containing Alexa Fluor® 488-HSE-DNA-bound Hsf1(1–408) trimers and the indicated components. One representative experiment of three is shown.

D  The HR-B proximal Hsc70 binding site is essential for Hsc70/DnaJB1-mediated dissociation of HSE-DNA-bound Hsf1. One representative experiment of three is shown.

Source data are available online for this figure.

exchanging subpopulation ($\leq$ 10% for 159–168 and 169–175; $\leq$ 17% for segment 176–189). In contrast, upon pre-incubation for 30 min at 25°C in the presence of Hsc70, DnaJB1, and ATP the high exchanging subpopulation increased to 20–30% (Fig 3D). In this case, the high exchanging subpopulation exchanged 67–91% of the amide protons, whereas the low exchanging subpopulation only exchanged 7–37% (Fig 3E), indicating that in the presence of Hsc70, DnaJB1, and ATP the helices of the leucine-zipper are unfolded.

## Binding sites adjacent to HR-B are essential for Hsc70-mediated Hsf1 monomerization

To elucidate whether the Hsc70 binding sites in the TAD are responsible for Hsc70-mediated dissociation of Hsf1 from DNA and also to assess any involvement of the previously suggested Hsc70 binding site, we deleted the entire TAD, residues 409–529. Hsf1(1–408) was dissociated from DNA by Hsc70 with similar kinetics as wild-type Hsf1 (Hsf1wt; Fig 4C), excluding any involvement of sites in the TAD in this process.

Consequently, we deleted the sites adjacent to the HR-B region, residues 202–213. This deletion in Hsf1 completely prevented dissociation of Hsf1 from DNA in the presence of Hsc70, DnaJB1, and ATP, indicating that these residues are essential for Hsc70-mediated Hsf1 monomerization (Fig 4D). Since a typical Hsc70 binding site consists of a core of up to five hydrophobic residues, preferably leucine, flanked by positively charged regions (Rüdiger et al, 1997), we exchanged the hydrophobic residues against serine, a small hydrophilic residue disfavoring Hsc70 binding, generating the two Hsf1 variants, Hsf1-I202S,L203S,V205S (Hsf1-202SSS) and Hsf1-I209S,L211S,L213S (Hsf1-209SSS). Hsc70-mediated dissociation of Hsf1 from DNA was either completely abrogated by these amino acid replacements or strongly inhibited (Fig 4D). Peptides harboring the same amino acid replacements were also not bound by Hsc70 (Fig 4B).

Taken together, these data indicate that Hsc70-mediated dissociation of Hsf1 from DNA requires binding of Hsc70 to a site adjacent to the trimerization domain. Hsc70 binding to the TAD may serve a different purpose.

## The number of available Hsc70 binding sites determines the rate of Hsf1 monomerization

We wondered whether binding of Hsc70 to one protomer of the Hsf1 trimer is sufficient for Hsc70-mediated monomerization. To address this question, we mixed monomeric Hsf1wt and Hsf1Δ(202–213) at different ratios and incubated the mixtures at 42°C for 10 min to form heterotrimers. Assuming that heterotrimers formed with the same probability as Hsf1wt and Hsf1Δ(202–213) homotrimers, different homo- and heterotrimeric species are formed according to a binomial distribution, which can be calculated using the formula $(a+b)^3$ for a 1:1 ratio, $(2a+b)^3$ for a ratio of 2:1, and $(a+2b)^3$ for a ratio of 1:2, whereby the coefficient of each element $a^3$, $a^2b$, $ab^2$, and $b^3$ normalized to the sum of all coefficients gives the relative abundance of the respective species. At a 2:1 ratio of Hsf1wt: Hsf1Δ(202–213), 29.6% of the Hsf1 trimers are expected to be Hsf1wt homotrimers and consequently have three Hsc70 binding sites available close to the trimerized region, 44% are expected to have two Hsf1wt protomers per Hsf1 trimer, 22.2% are expected to have a single Hsf1wt per Hsf1 trimer, and 3.7% are expected Hsf1Δ(202–213) homotrimers. The fraction of the different homo- and heterotrimers change at the different mixing ratios as indicated in Fig 5A. When we subjected these mixtures to Hsc70-mediated dissociation from DNA, we observed that the rate of dissociation decreased significantly with decreasing Hsf1wt:Hsf1Δ(202–213) ratios (Figs 5A and EV2B). Assuming that binding of a single Hsc70 is sufficient for Hsf1 monomerization and thus dissociation from DNA at wild-type rates and that only the Hsf1Δ(202–213)

homotrimer cannot be dissociated from DNA, as shown above, the overall rate of dissociation should not change for the different mixing ratios. Only the total amplitude of the dissociation reaction should change, because the Hsf1Δ(202–213) homotrimers would remain bound to a fraction of the DNA and thus cause a residual average polarization higher than the polarization of unbound DNA. However, using this model, the simulated curves did not fit our experimental data (dashed lines in Fig EV2A). The alternative hypothesis that always three Hsc70 binding sites are necessary would lead to a similar situation, just with smaller amplitudes representing the small fraction of Hsf1wt homotrimers, also not explaining our experimental data. We concluded that the rate of Hsf1 monomerization depends on the number of Hsc70s bound simultaneously or sequentially to different protomers of the Hsf1 trimer. An equation describing this situation fitted our experimental data reasonably well (see Appendix, Fig EV2C). Binding of a single Hsc70 per Hsf1 trimer is able to monomerize the Hsf1 trimer albeit at a very low rate, whereas action of Hsc70 on two protomers of the Hsf1 trimer more efficiently monomerizes Hsf1 trimers, and action of Hsc70 on all three protomers within the Hsf1 trimer is still more efficient in monomerization (Fig EV2D).

## Hsc70 monomerizes Hsf1 by entropic pulling

The concept of entropic pulling was proposed for Hsp70 action during import of polypeptides into mitochondria and for protein disaggregation (De Los Rios et al, 2006). Briefly, Hsp70 binds to incoming polypeptides close to the membrane. Due to the excluded volume of the bulky Hsp70, the conformational freedom of the polypeptide is limited and thus the entropy low. Movement of the peptide into the mitochondrial matrix increases the distance of the polypeptide-bound Hsp70 to the membrane, allowing for more conformational freedom of the polypeptide and thus increases the entropy. Since chemical reaction can be driven by increase in entropy as well as decrease in enthalpy, a force is generated that pulls the polypeptide into the mitochondrial matrix. De Los Rios et al calculated a pulling force of around 10–20 pN that decrease with increasing length of the incoming polypeptide and will reach 0 pN once about 30 residues are imported. To drive further import, a new Hsp70 needs to bind to the incoming polypeptide close to the membrane.

To test this hypothesis, we moved the Hsc70 binding site away from the HR-B region along the intrinsically disordered regulatory domain. Already when the Hsc70 binding site is 10 residues away from HR-B, Hsc70 dissociated Hsf1 from DNA with a significantly lower rate (Fig 5B). At a distance of 20 residues, Hsc70 was not anymore able to dissociate Hsf1 from the DNA, indicating that monomerization was not anymore possible. These results suggest that Hsc70 monomerizes Hsf1 trimers by entropic pulling.

To substantiate this hypothesis, we tested whether simple binding of an antibody close to HR-B would be sufficient to unzip the leucine-zipper of Hsf1. We inserted a FLAG epitope between HR-B and the Hsc70 binding site or 10 and 20 residues downstream of HR-B. We treated anti-FLAG antibodies with DTT to split them in half (Appendix Fig S3) and added them to DNA-bound FLAG epitope containing Hsf1 in the absence of Hsc70 and DnaJB1. Surprisingly, we did not observe any dissociation of Hsf1 (Fig 5C). This was not due to a failure of the FLAG antibody halfmers to bind

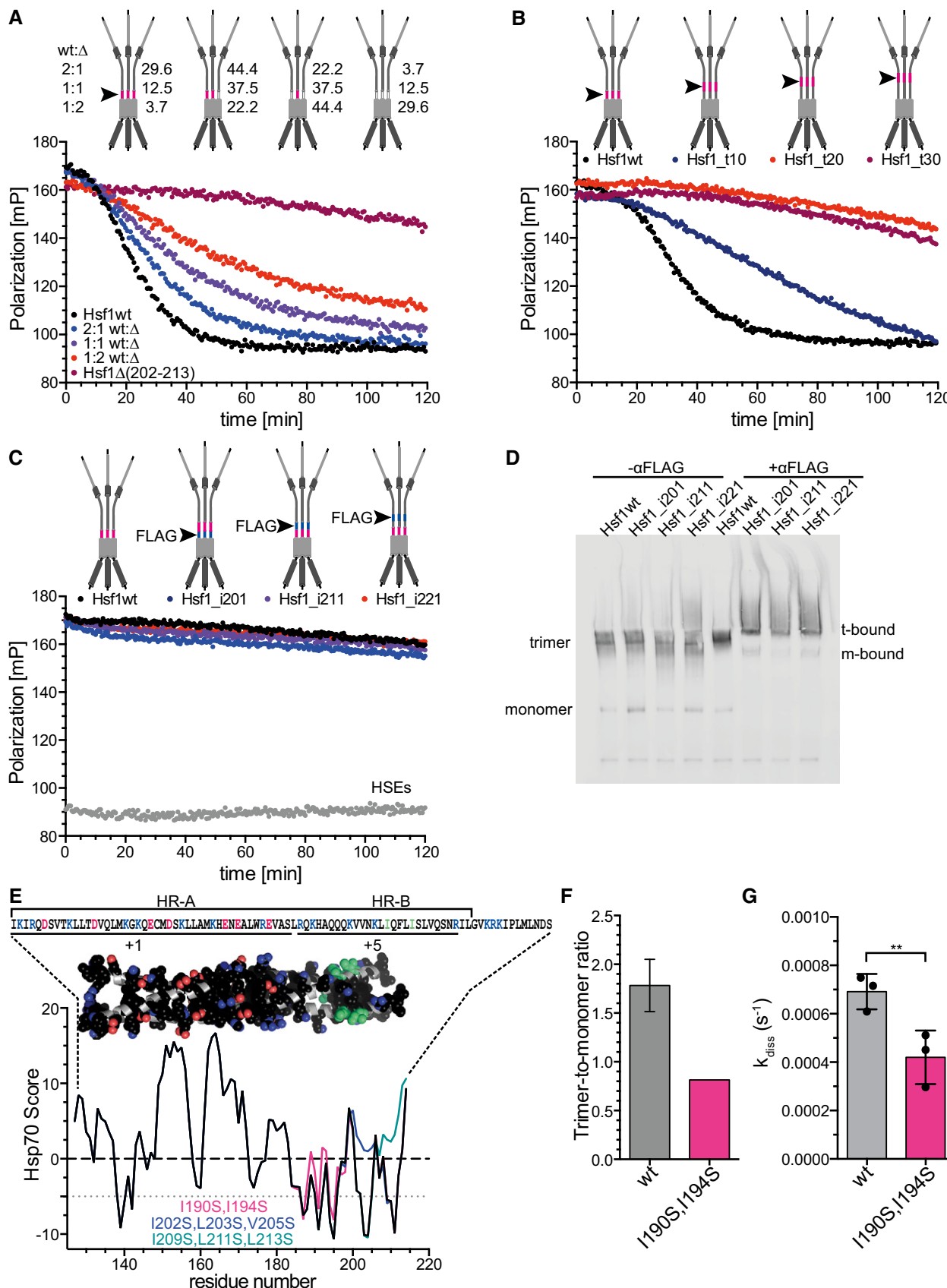

**Figure 5.**

◀ **Figure 5.  Hsc70/DnaJB1 monomerize Hsf1 by cycles of successive entropic pulling.**

A   The rate of Hsc70/DnaJB1-mediated dissociation of HSE-DNA-bound Hsf1 depends on the number of HR-B proximal Hsc70 binding sites in the Hsf1 trimer. Hsf1wt (wt) and Hsf1Δ(202–213) (Δ) monomers were mixed in the indicated ratios and heat shocked at 42°C for 10 min to form mixtures of different homo- and heterotrimers as indicated with the cartoons (the red line and the arrowhead indicate the Hsc70 binding site). Numbers indicate the relative fraction of the different species. For fit of the data, see Fig EV2.

B   Moving the Hsc70 binding site away from the trimerization domain reduces the rate of Hsc70/DnaJB1-mediated dissociation of HSE-DNA-bound Hsf1. Hsf1_t10/20/30, region 202–213 moved by 10, 20, or 30 residues toward the C-terminus.

C   Anti-FLAG antibodies are not able to dissociate HSE-DNA-bound Hsf1. Red lines in the cartoon indicate the Hsc70 binding site; blue lines and arrowhead indicate the inserted FLAG epitope DYKDDDDK. Hsf1_i201/211/221, FLAG epitope inserted after residue 201, 211, or 221. Anti-FLAG antibodies were split in halfmers by incubation with 2 mM DTT (see Appendix Fig S3).

D   Anti-FLAG antibody halfmers bound to FLAG epitope containing Hsf1 variants. Hsf1wt and FLAG-insertion variants were analyzed by blue-native gel electrophoresis in the absence or presence of DTT-treated anti-FLAG antibodies as indicated and Hsf1 detected by immunoblotting; t-bound, FLAG antibody-bound Hsf1 trimers; m-bound, FLAG antibody-bound Hsf1 monomers.

E   Predicted Hsc70 binding sites in the trimerization domain of Hsf1. Hsp70 score of the trimerization region residues 130–216 of Hsf1wt, black; Hsf1-I190S,I194S, magenta; Hsf1-I202S,L203S,V205S, blue; and Hsf1-I209S,L211S,L213S, green. Values below −5 are considered good Hsc70 binding sites. Above the graph is the trimeric homology model of the trimerization domain residues 130–203 with side chains of hydrophobic and charged residues in space-filling representation in atom colors with carbon in black except for Ile190 and Ile194 where carbon is shown in green. Above the model is the corresponding sequence. Positively charged residues, blue; negatively charged residues, red. Lines below the sequence indicate two distinct regions with net charge +1 and +5.

F   Trimer-to-monomer ratio of freshly purified Hsf1wt and Hsf1-I190S,I194S in the absence of heat shock, determined by gel filtration and BN-PAGE (see Appendix Fig S4). Hsf1wt, mean ± SD (n = 3).

G   Rate of Hsc70/DnaJB1-mediated dissociation of HSE-DNA-bound Hsf1wt and Hsf1-I190S,I194S; mean ± SD (n = 3); **, P < 0.01; (paired t-test).

Source data are available online for this figure.

---

to the FLAG epitope containing Hsf1 trimers as demonstrated by BN-PAGE followed by Western blot (Fig 5D).

We hypothesized that pulling from a single site at the end of the trimerization domain may not be sufficient to unzip the entire domain, since the trimerization domain has a length of 75 residues and the entropic pulling force failed already when Hsc70 bound more than 20 residues away from the leucine-zipper. Close inspection of the HR-A/B region revealed that the sequence contains a large number of hydrophobic residues, as expected for a leucine-zipper, but unexpectedly the C-terminal part of the zipper (HR-B) contains 5 positively charged residues, which favor Hsc70 binding, and not a single negatively charged residue, which disfavor Hsc70 binding. Thus, this region of the trimerization domain contains several potential Hsc70 binding sites, as also evident from the Hsc70 binding site prediction (Fig 5E). To compromise Hsc70 binding in this region is rather difficult, since replacing hydrophobic residues could disturb the leucine-zipper and prevent Hsf1 trimerization altogether. We used a model of the trimerization domain kindly provided by A. Bracher (Neudegger *et al*, 2016) which ended at residue 182 and used the iTASSER homology modeling software (https://zhanglab.ccmb. med.umich.edu/I-TASSER/; Zhang, 2008; Roy *et al*, 2010) to extend the model to residue 203. In this model, we discovered two isoleucine residues (I190 and I194) that did not point toward the zipper interface. Replacing these two isoleucines by serines only moderately reduced the propensity of this region to bind to Hsc70 (magenta line in Fig 5E) as compared to the more drastic changes introduced by replacing I202, L203, and V205 (blue line) or I209, L211, and L213 (green line) by serines. Surprisingly, when we purified Hsf1-I190S,I194S we retrieved by gel filtration significantly less trimers and more monomers as compared to Hsf1wt (Fig 5F, Appendix Fig S4), suggesting that the amino acid replacements have destabilized the trimeric state. Nevertheless, Hsc70-mediated monomerization and thus dissociation from DNA occurred at significantly lower rate for Hsf1-I190S,I194S than for Hsf1wt (Fig 5G). These data suggest that binding of Hsc70 to this

region (190–194) located within HR-B contributes to Hsf1 monomerization. Altogether our results suggest that Hsc70 monomerizes Hsf1 by successive entropic pulling unzipping the leucine-zipper step by step.

**Amino acid replacements in Hsc70-binding sites potentiate heat shock reporter expression**

To test the consequences of our findings in a cell culture model system, we stably transfected HSF1$^{-/-}$ mouse embryonic fibroblasts (MEFs) with a heat shock reporter expressing mTagBFP under the control of the HSPA6 promoter. We then transiently transfected these cells with plasmids expressing wild-type or mutant Hsf1 and hrGFP in an artificial operon using an internal ribosomal entry site (IRES) (Fig 6A). We prepared lysates of transfected cells and analyzed the amount of wild-type and mutant Hsf1 by SDS–PAGE and immunoblotting, the amount of trimeric Hsf1 by blue-native PAGE and immunoblotting, and the ability of the expressed Hsf1 to bind to DNA by electrophoretic mobility shift assay. We did not detect significant differences in overall protein amount between wild-type and mutant Hsf1 proteins in the lysates of the transfected MEFs (Fig EV3A and B). In contrast, higher amounts of trimeric Hsf1 were detected for variants with mutations in the HR-B proximal Hsc70 binding site (Fig EV3C and D). We also observed a band shift of fluorescent-labeled HSE-DNA in the presence of Hsf1 containing MEF lysate but not in lysate from cells transfected with the empty vector (Fig EV3E and F). The increased amount of trimeric Hsf1 in the lysate of the MEFs was not due to a lower temperature transition of the Hsf1 mutant variants, as none of the Hsf1 variants with replacements in the HR-B proximal Hsc70 binding site had a thermal transition temperature lower than Hsf1wt (Fig EV3G and H). In fact, the transition temperature of Hsf1Δ(202–213) was increased by 1.7°C.

Analysis of the transfected cells by flow cytometry revealed that the GFP-positive cells were not a separate subpopulation distinguishing transfected from non-transfected cells, but GFP

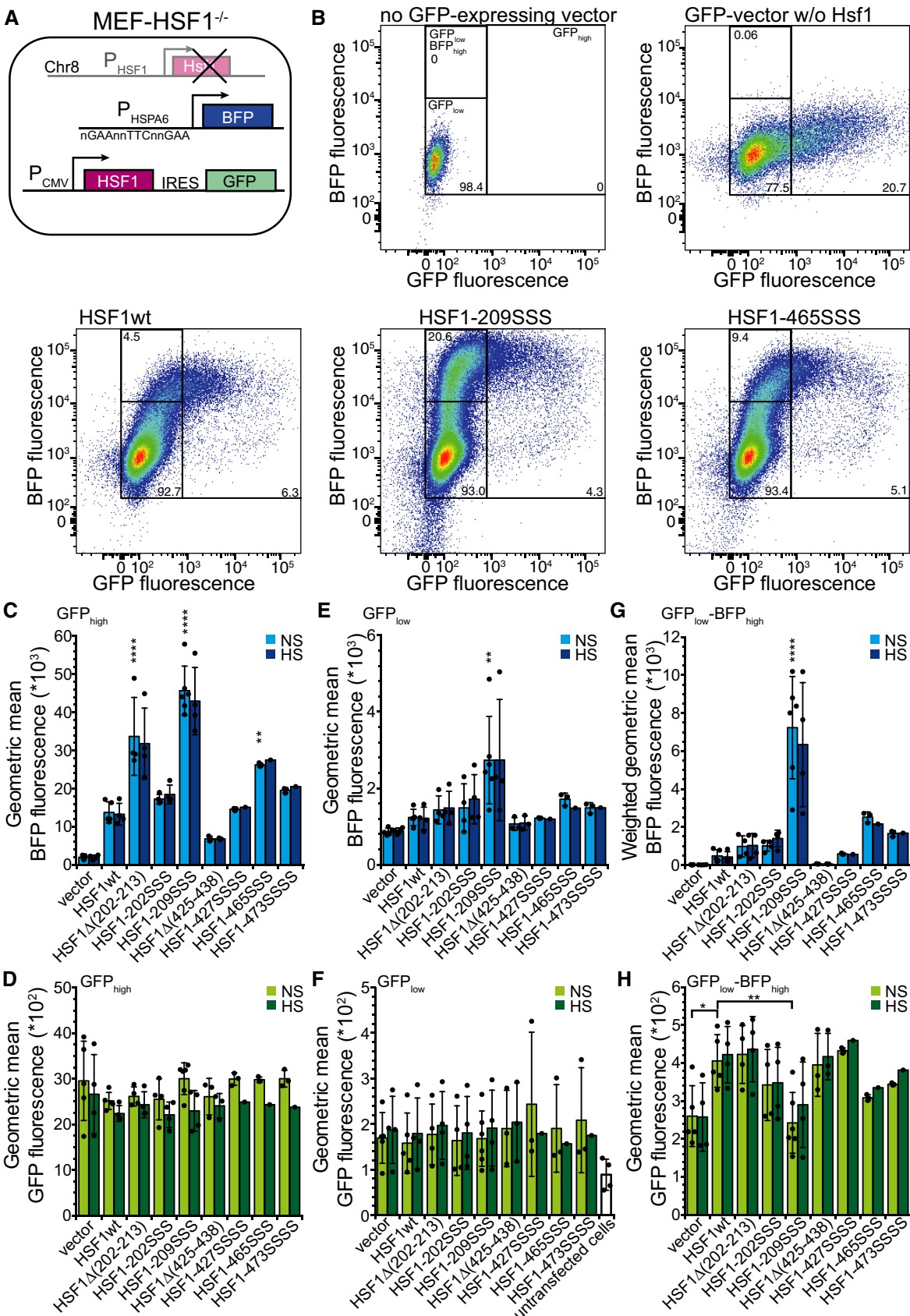

**Figure 6.**

◀  **Figure 6.  Compromising Hsc70 binding potentiates expression of a heat shock reporter in HSF1$^{-/-}$ mouse embryonic fibroblasts (MEF).**

A   Schematics of the used system. MEFs, in which HSF1 on chromosome 8 was deleted, were stably transfected with mTagBFP expressed under the control of the HSPA6 promoter and subsequently transiently transfected with plasmids that expressed HSF1wt or mutant variants in an artificial operon with hrGFP, which is translated from an IRES element.

B   Exemplary flow cytometry data plotting BFP fluorescence versus GFP fluorescence, indicating the gates used for analysis. Numbers indicate the fraction (%) of cells (forward scatter singlets) in each gate. Color indicates the cell density from dark blue (low) to red (high).

C–H   Geometric mean of BFP (C, E) and GFP fluorescence (D, F, H) of cells grown at 37°C (non-stress, NS) or incubated for 1 h at 43°C with a recovery period of 3 h at 37°C (heat shocked, HS) in the GFP$_{high}$ (C, D), GFP$_{low}$ (E, F), and GFP$_{low}$-BFP$_{high}$ (H) gates. (G) Geometric mean of BFP fluorescence weighted by the relative number of cells in the GFP$_{low}$-BFP$_{high}$ gate (fluorescence * fraction forward scatter singlets). Note that the GFP$_{low}$ gate also comprises the GFP$_{low}$-BFP$_{high}$ gate. Hsf1-202SSS, Hsf1-I202S,L203S,V205S; Hsf1-209SSS, Hsf1-I209S,L211S,L213S; Hsf1-427SSSS, Hsf1-M427L,L429S,L432S,L436S; Hsf1-465SSS, Hsf1-L465S,V466S,Y468S; Hsf1-473SSSS, Hsf1-L473S,F474S,L475S,L476S; geometric mean ± SD of up to six independent experiments; ANOVA Sidak's multiple comparison, *$P < 0.05$; **$P < 0.01$; ****$P < 0.0001$.

fluorescence of cells was distributed over three orders of magnitude from 84 relative fluorescence units (RFU), the average level of non-transfected cells (Fig 6B top left panel), up to $10^5$ RFU (Fig 6B, top right panel). According to the instrument manufacturer, the fluorescence detection is linear over the entire range and the amount of GFP in cells correlates linearly with the detected fluorescence. We conclude that the relative amount of GFP in the transfected cells varies over three orders of magnitude. This was independent of the transfection method as liposome and artificial polymer-based transfection and electroporation were used with similar results, and it was independent of the cell type and also found in HeLa cells (see below). Since Hsf1 and GFP were expressed from an artificial operon and translated from the same mRNA, GFP fluorescence can be taken as a proxy for Hsf1 levels and we infer that the Hsf1 concentration in the transfected cells also varies over such a wide range. This is important to note as Hsf1 trimerization is concentration-dependent (Hentze *et al*, 2016).

For the vector control, BFP reporter fluorescence did not significantly increase with increasing GFP fluorescence (Fig 6B top right panel), whereas for cells expressing Hsf1wt BFP fluorescence increased linearly with GFP fluorescence in the GFP$_{low}$ gate but stayed constant at GFP fluorescence values $> 10^3$ (GFP$_{high}$ gate) (Fig 6B lower left panel). These data indicate that increasing Hsf1 concentrations leads to increasing BFP fluorescence even in the absence of heat shock presumably due to concentration-dependent Hsf1 trimerization. At very high Hsf1 concentrations, BFP fluorescence did not further increase, suggesting saturation of heat shock gene transcription.

Cells in the GFP$_{high}$ gate expressing Hsf1 variants mutated in the HR-B proximal Hsc70 binding site (Hsf1Δ(202–213) and Hsf1-209SSS) exhibited an up to threefold higher BFP fluorescence than Hsf1wt-expressing cells and cells expressing Hsf1 variants mutated in the here newly discovered Hsc70 binding sites in the TAD, Hsf1-465SSS and Hsf1-473SSSS, a roughly twofold and 1.4-fold increased BFP fluorescence (Fig 6B and C), demonstrating that interfering with Hsc70 binding to Hsf1 in cells potentiates heat shock gene transcription. In contrast, deletion or mutation of the previously proposed Hsc70 binding site in the TAD (Hsf1Δ(425–438) and Hsf1-427SSSS) did not increase BFP fluorescence. Surprisingly, Hsf1-202SSS that could not be monomerized by Hsc70 *in vitro* did not lead to a strongly increased BFP fluorescence. Even more surprisingly, deletion of residues 202–213 increased BFP fluorescence less prominently than replacing residues 209, 211, and 213 by serine (see below Discussion). The increased BFP fluorescence could not be due to increased expression of Hsf1 mutants as the

mean GFP fluorescence was not significantly different between Hsf1wt and Hsf1 mutant expressing cells and thus differences in Hsf1 levels are unlikely (Fig 6D, see also Fig EV3A and B).

A 43°C heat shock followed by 3-h recovery at 37°C did not lead to a further increase in BFP fluorescence in the GFP$_{high}$ cells (Fig 6C dark blue bars). There could be two reasons for this observation. High levels of Hsf1 trimerize spontaneously in the absence of heat shock, increasing the levels of heat shock proteins sufficiently to overcome the heat shock without further induction of the HSR. Alternatively or in addition, transfection itself is a stress, inducing the HSR and leaves the MEFs refractory to a second stress. To investigate the first hypothesis, we analyzed the cells in the GFP$_{low}$ gate that exhibited a mean GFP fluorescence of 171 (Fig 6F average over all green bars) as compared to 84 of non-transfected cells (Fig 6F white bar) and 2,584 of GFP$_{high}$ cells (Fig 6D average over all green bars), suggesting an average 29-fold difference in GFP and thus presumably Hsf1 expression between cells in the GFP$_{low}$ and the cells in the GFP$_{high}$ gates (baseline fluorescence subtracted). The mean BFP fluorescence was significantly increased only for Hsf1-209SSS-expressing cells (Fig 6E). However, most cells in this gate are non-transfected since MEFs are particularly difficult to transfect and transfection efficiency is low. To exclude the non-transfected cells from the analysis, we focused on cells with low GFP fluorescence but high BFP fluorescence (GFP$_{low}$-BFP$_{high}$ gate; Fig 6B top left panel). Very few cells transfected with the vector control were found within this gate (Fig 6B top right panel), but on average 2.2 ± 1.7% of all Hsf1wt and 19.1 ± 7.5% of Hsf1-209SSS-expressing cells. In addition, the mean BFP fluorescence was increased for Hsf1-209SSS-expressing cells twofold over Hsf1wt-expressing cells within this gate. To account for both differences, relative number of cells in the GFP$_{low}$-BFP$_{high}$ gate and mean BFP fluorescence were multiplied (weighted geometric mean BFP fluorescence; Fig 6G) revealing a 17-fold difference between Hsf1wt- and Hsf1-209SSS-expressing cells, although according to GFP fluorescence Hsf1-209SSS is expressed to a lower level than Hsf1wt (Fig 6H, 232 versus 401 RFU). These data indicate that interfering with Hsc70-mediated monomerization of Hsf1 strongly increases heat shock gene transcription, which becomes especially apparent at low Hsf1 expression. As observed for the GFP$_{high}$ gate, cells expressing Hsf1-202SSS or Hsf1Δ(202–213) did not show a comparable increase in BFP fluorescence (2.5- and 2.2-fold relative to Hsf1wt). Again, heat shock did not increase BFP reporter activity even so GFP fluorescence was significantly lower than in the GFP$_{high}$ cells. This suggests that stress caused by transfection might have triggered transcription of BFP but also transcription of the

HSPA1A/B and HSPA8 genes, increasing Hsp70 and Hsc70 levels. These high BFP and Hsp70/Hsc70 levels may remain high due to the low turnover rate of BFP and Hsp70/Hsc70 and the low division rate of these cells. High Hsp70/Hsc70 concentrations would suppress further increase in heat shock gene transcription.

To analyze the effects of our Hsf1 variants on the heat shock response in a more robust cell line, we stably transfected HeLa cells with the BFP heat shock transcription reporter and treated and analyzed the cells in a similar way as the MEFs (Fig 7A). Since HeLa cells contain endogenous Hsf1, heat-inducible BFP fluorescence was observed in the empty vector control independent of GFP fluorescence (Fig 7B top right panels HS versus NS; and Fig 7C; 4.6-fold increase). Expression of Hsf1wt in these cells increased the BFP fluorescence 7.5-fold in the GFP$_{high}$ gate, which is higher than heat induction of the empty vector control (Fig 7C). In the GFP$_{high}$ cells, we again observed a strong increase in BFP fluorescence for the Hsf1 variants mutated in the Hsc70 binding sites adjacent to the trimerization domain (6.6- and 6.9-fold for Hsf1-209SSS relative to Hsf1wt-expressing control and heat shocked cells, respectively, Fig 7B and C). Similarly, increased BFP fluorescence was also observed for Hsf1 variants with altered Hsc70 binding site in the TAD (2.2- and 2.7-fold for Hsf1-465SSS relative to Hsf1wt-expressing control and heat shocked cells, respectively; Figs 7B and C, and EV4A). For most of the Hsf1 constructs, we observed a slight heat shock induction, albeit not statistically significant. However, this changed when we analyzed the cells in the GFP$_{low}$ gate (low levels of Hsf1) that had an average GFP fluorescence of 72 (Fig 7F, average over all green bars) as compared to 34 (Fig 7F white bar) for non-transfected cells and 1,026 for the GFP$_{high}$ cells (Fig 7D average over green bars excluding the vector control; 26-fold difference in the geometric mean). For each Hsf1 construct, we observed a highly significant heat shock induction (Fig 7E; only indicated for the vector control). In addition, BFP fluorescence of unstressed cells harboring Hsf1-209SSS was significantly increased as compared to Hsf1wt-containing unstressed cells (Fig 7E, light blue bars), and heat shock-induced BFP fluorescence was significantly increased for Hsf1-202SSS and Hsf1-209SSS (Fig 7E, dark blue bars).

To minimize the contribution of non-transfected cells and endogenous Hsf1, we selected the nested gates GFP$_{low}$-BFP$_{high}$37 and GFP$_{low}$-BFP$_{high}$43 (Fig 7B, top left panel), which captured no or very few cells from the vector control (Fig 7B top right panels), for analysis of control (GFP$_{low}$-BFP$_{high}$37) and heat shocked cells (GFP$_{low}$-BFP$_{high}$43). For both gates, the number of cells and the geometric mean of BFP fluorescence were strongly increased for Hsf1-209SSS- and Hsf1-465SSS-containing cells (Figs 7G and H, and EV4B and C), indicating that Hsc70-mediated negative feedback loop is impaired.

## Discussion

In this study, we gained several important insights into the regulation of the HSR. We demonstrate that Hsc70 together with its J-domain co-chaperone DnaJB1 removes Hsf1 from heat shock promoter DNA by monomerizing the Hsf1 trimers. Thus, the HSR can be shut off without Hsf1 acetylation or degradation and Hsf1 can be recycled. Of note, we do not exclude that J-domain proteins other than DnaJB1 or nucleotide exchange factors other than Apg2

could also cooperate in Hsc70-mediated monomerization of Hsf1. Hsf1 monomerization starts from an Hsc70 binding site C-terminal of the trimerization domain proximal to HR-B and proceeds toward the N-terminus of the trimerization domain through stepwise unzipping the leucine-zipper by entropic pulling (Fig 8A). We show that starting this repeated binding of Hsc70 at several protomers of the Hsf1 trimer allows for more rapid disassembly. This mechanism makes the biggest contribution to attenuation of heat shock gene transcription in cell culture models. We also found that binding of Hsc70 to a newly identified site in the TAD contributes to attenuation in the cell culture models, most likely by interfering with the Hsf1-mediated release of stalled RNA polymerase. It is interesting that this binding site has the highest affinity for Hsc70 ($K_D$ ca. 5 μM), indicating that increasing concentrations of free Hsc70 during heat shock transcription will on average first bind to this site before binding to the HR-B proximal site.

The flow cytometry experiments reveal several insights into the system in a cellular context. Increasing expression of Hsf1 (increasing GFP fluorescence) led to increasing transcription of the BFP reporter (increasing fluorescence) and should also increase transcription of endogenous heat shock genes including transcription of HSPA1A and HSPA1B encoding Hsp70 and HSPA8 encoding Hsc70. This is consistent with the earlier finding that Hsf1 trimerization is concentration-dependent (Hentze *et al*, 2016). In turn, increasing Hsp70/Hsc70 concentrations will more rapidly monomerize trimeric Hsf1, attenuating transcription and reestablishing the heat shock response at a new equilibrium. It was shown earlier that high levels of Hsp70 inhibit the induction of the HSR (Baler *et al*, 1996). Such an equilibrium, here induced by Hsf1 overexpression, is conceptionally similar to the equilibrium reached during the attenuation phase of the HSR when high temperatures continuously drive Hsf1 trimerization due to the thermosensory function of Hsf1 (Hentze *et al*, 2016) and high levels of Hsp70, resulting from Hsf1-induced transcription, continuously monomerize trimeric Hsf1. At the highest Hsf1 concentrations (GFP$_{high}$) gate, the BFP fluorescence leveled off, indicating that another mechanism independent of Hsf1 concentration must limit heat shock gene transcription. This maximum is still dependent on interaction with Hsp70/Hsc70 close to the trimerization domain and in the TAD as cells containing Hsf1 variants with impaired interaction with Hsc70 exhibited an even higher BFP fluorescence at high Hsf1 concentrations as compared to Hsf1wt. The maximum transcriptional activity of Hsf1-209SSS might be limited by interaction of its TAD with Hsc70.

Hsf1-202SSS induced the heat shock reporter not to the same extent as Hsf1-209SSS, although *in vitro* it was not dissociated from DNA and monomerized by Hsc70 and DnaJB1. Even more surprising was the finding that deletion of the entire HR-B proximal Hsc70-binding site from 202 to 213 (Hsf1Δ(202–213)), comprising the region mutated in Hsf1-209SSS (I209S,L211S,L213S), had a much weaker effect on BFP reporter induction than Hsf1-209SSS (Figs 6C, E and G, and 7C, E and G). Both results together suggest that the region from 202 to 208 is important for the transcriptional activity of Hsf1, and mutation or deletion of this region dampens heat shock gene induction. The precise mechanism of this effect requires further investigations. In HeLa cells at very high GFP fluorescence, a subpopulation appears with rather lower BFP fluorescence. Back tracing indicates that these cells are not distinguishable in forward or side scatter from the other cells. It is not clear what attenuates

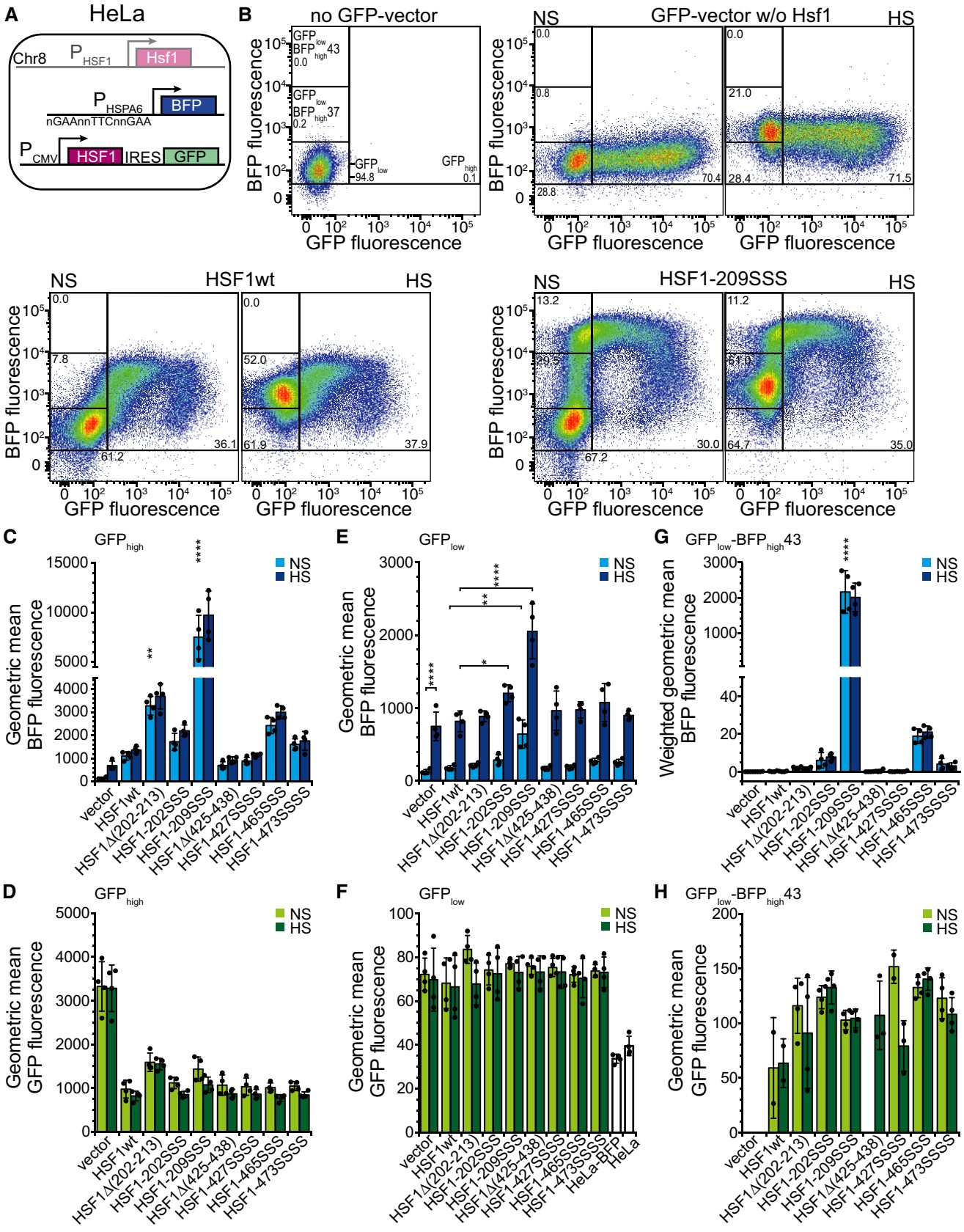

**Figure 7.**

◀

**Figure 7. Compromising Hsc70 binding potentiates expression of a heat shock reporter under non-stress and heat shock conditions in HeLa cells.**

A   Schematics of the used system. HeLa cells were stably transfected with mTagBFP expressed under the control of the HSPA6 promoter and subsequently transiently transfected with plasmids that expressed HSF1wt or mutant variants in an artificial operon with hrGFP, which is translated from an IRES element.

B   Exemplary flow cytometry data plotting BFP fluorescence versus GFP fluorescence, indicating the gates used for analysis. Numbers indicate the fraction (%) of cells (forward scatter singlets) in each gate. Color indicates the cell density from dark blue (low) to red (high). NS, cells grown at 37°C; HS, cells heat shocked at 43°C for 1 h with a recovery period of 3 h at 37°C.

C–H  Geometric mean of BFP (C, E) and GFP fluorescence (D, F, H) in the GFP$_{high}$ (C, D), GFP$_{low}$ (E, F), and GFP$_{low}$-BFP$_{high}$43 (H) gates. Of note, the number of data points shows the number of experiments in which a significant number of cells was found in this gate and missing bars indicate that cells did not fall within this gate in more than two out of four experiments. (G) Geometric mean of BFP fluorescence weighted by the relative number of cells in the GFP$_{low}$-BFP$_{high}$43 gate (fluorescence * fraction forward scatter singlets). Note that the GFP$_{low}$ gate also comprises the GFP$_{low}$-BFP$_{high}$37 and that the GFP$_{low}$-BFP$_{high}$37 gate also comprises the GFP$_{low}$-BFP$_{high}$43 gate. Hsf1-202SSS, Hsf1-I202S,L203S,V205S; Hsf1-209SSS, Hsf1-I209S,L211S,L213S; Hsf1-427SSSS, Hsf1-M427S,L429S,L432S,L436S; Hsf1-465SSS, Hsf1-L465S,V466S,Y468S; Hsf1-473SSSS, Hsf1-L473S,F474S,L475S,L476S; geometric mean ± SD of 4 independent experiments; ANOVA Sidak's multiple comparison, *$P < 0.05$; **$P < 0.01$; ****$P < 0.0001$.

transcription in these cells but this might indicate that too high Hsf1 activity is detrimental for cell health. Consistently, in some transfection experiments using the Hsf1-209SSS construct we observed increased cell death.

One might wonder why the affinity of Hsc70 for the critical HR-B proximal site is so low ($K_D$ ca. 30 μM). Firstly, it is important to understand that the equilibrium dissociation constant determined for the ADP-bound state only insufficiently describes the real affinity of Hsc70 for a binding site. In the presence of ATP and a J-domain protein, Hsc70·ATP associates with high rates with the protein client, and then, the J-domain protein in synergism with the bound substrate polypeptide stimulates ATP hydrolysis in Hsc70, leading to transition to the so-called high-affinity state with low substrate dissociation rates. This targeting mechanism leads to a non-equilibrium situation that decreases the apparent $K_D$ by several orders of magnitude and was coined ultra-affinity (De Los Rios & Barducci, 2014). The actual affinity of Hsc70 to this site depends on the local concentration of DnaJB1, which seems to have also some affinity for these sites to increase local concentration and the concentration of the nucleotide exchange factor (Fig 1I) that allows for ADP release, ATP rebinding, and substrate release. Secondly, the cellular concentration of Hsc70 averaged over 11 different cancer cell lines is around 12–18 μM under non-stress conditions (Geiger *et al*, 2012) assuming a total cellular protein concentration of 100–150 mg/ml (Gillen & Forbush, 1999; Finka *et al*, 2015) and can reach 14–21 μM upon a mild heat shock of 41°C for 4 h (Finka *et al*, 2015).

In contrast, human Hsp70 (HSPA1A/B) is barely detectable in most non-cancer cells and reaches only about 0.04–0.06 μM upon heat shock (Finka *et al*, 2015), but can reach around 0.7–10 μM in cancer cells (Geiger *et al*, 2012). It is very likely that the HR-B proximal site evolved for such an affinity to Hsc70 to allow for a high enough concentration of Hsc70 in the cell. If the affinity of this site for Hsc70 was higher ($K_D$ lower), Hsc70 might disassemble Hsf1 already at lower concentrations when the cell still needs more Hsc70 and other stress proteins.

This also explains why the reaction is so exquisitely sensitive to the concentration of Hsc70 and DnaJB1 and why high concentrations of the nucleotide exchange factor Apg2 inhibit the reaction. Apg2 accelerates ADP dissociation and thus rebinding of ATP and release of Hsc70 from Hsf1. If the first Hsc70 is released from Hsf1 before another Hsc70 can bind to the next Hsc70 binding site that becomes transiently accessible through entropic pulling at the first site, unzipping cannot be efficient. Our data also demonstrate that working on several protomers of the Hsf1 trimer more efficiently disassembles Hsf1 trimers than pulling on a single protomer,

providing an additional explanation for the necessity of higher Hsc70 concentrations. A reasonable explanation for this observation is that the entropic pulling force is larger, if several Hsc70 molecules are bound in close proximity to each other to individual protomers of the Hsf1 trimer, increasing local crowding and thereby decreasing the conformational freedom of the HR-B proximal region of the intrinsically disordered regulatory domain.

It is interesting that inserting a FLAG epitope before or after the HR-B proximal Hsc70 binding site and using anti-FLAG antibody halfmers did not lead to disassembly of Hsf1 trimers. This observation contrasts disassembly of clathrin coats that was efficiently performed by replacing the Hsc70 binding site and using anti-FLAG Fab fragments (Sousa *et al*, 2016), indicating that a single pull at all three Hsf1 protomers in the Hsf1 trimer is not sufficient for disassembly. It would not be possible to add more FLAG epitopes along the trimerization domain because they would disrupt the triple leucine-zipper. Conceptionally, disassembly of Hsf1 trimers is more similar to protein translocation through a membrane and to disaggregation. In both cases, successive binding events along a polypeptide chain pull the protein through the membrane pore and out of the aggregate, respectively. In the case of clathrin-coat disassembly, it was hypothesized that Hsp70 acts more through a collision pressure mechanism than by entropic pulling as it works from the inside of the cages not from the outside (Sousa *et al*, 2016). To our knowledge, the entropic pulling hypothesis has not been rigorously tested in the context of protein translocation or protein disaggregation and our results presented here constitute the first rigorous test of this hypothesis for the mode of Hsp70 action.

Based on the here presented data and published literature, we propose the following model for the regulation of the HSR (Fig 8B). In unstressed cells, Hsf1 is in a monomer–dimer equilibrium occasionally trimerizing in a concentration- and Hsp90-dependent manner and binding to HSEs in the genome. Hsc70 may bind to the TAD to attenuate heat shock gene transcription and to the HR-B proximal Hsc70 binding site to disassemble either free Hsf1 trimers before DNA binding (not shown in Fig 8B) or DNA-bound Hsf1 trimers into monomers. This situation leads to a low constitutive transcription of heat shock genes. Upon heat shock, Hsf1 trimerization is accelerated by Hsf1 thermosensing, integrating over temperature and time spent at elevated temperatures (Hentze *et al*, 2016). At the same time, Hsf1 monomerization is slowed down, due to titration of Hsc70 to misfolded and aggregated proteins. Thus, heat shock gene transcription increases rapidly and Hsc70 and Hsp70 concentrations increase. As soon as the concentration of free Hsp70/Hsc70 is sufficient, Hsc70 or Hsp70 binds to the TAD of Hsf1, the site with the

highest affinity, reducing transcriptional activation. This interaction is transient following the known Hsp70 ATPase cycle and depends on the nucleotide exchange factor concentration. As Hsc70 and Hsp70 concentrations continue to increase, Hsp70/Hsc70 concentrations will be sufficient to bind to the HR-B site and to disassemble the Hsf1 trimer and thus removing it from DNA, reducing heat shock gene transcription to a level that keeps Hsc70 and Hsp70 at the necessary concentrations. This Hsf1 activation/attenuation cycle is

highly sensitive to the concentration of free Hsc70. Thus, Hsf1 is a sensor for the cellular levels of free Hsc70 and Hsp70. Our model explains a host of literature accumulated over the last 40 years. For example, in HeLa cells more Hsp70 was bound to Hsf1 during attenuation and recovery phase then before heat shock, and the binding site we identified in the TAD explains why transient overexpression of Hsp70 or DnaJB1/Hdj1 expressing genes reduced transcription of a fusion construct between the TAD of Hsf1 and the DBD of Gal4

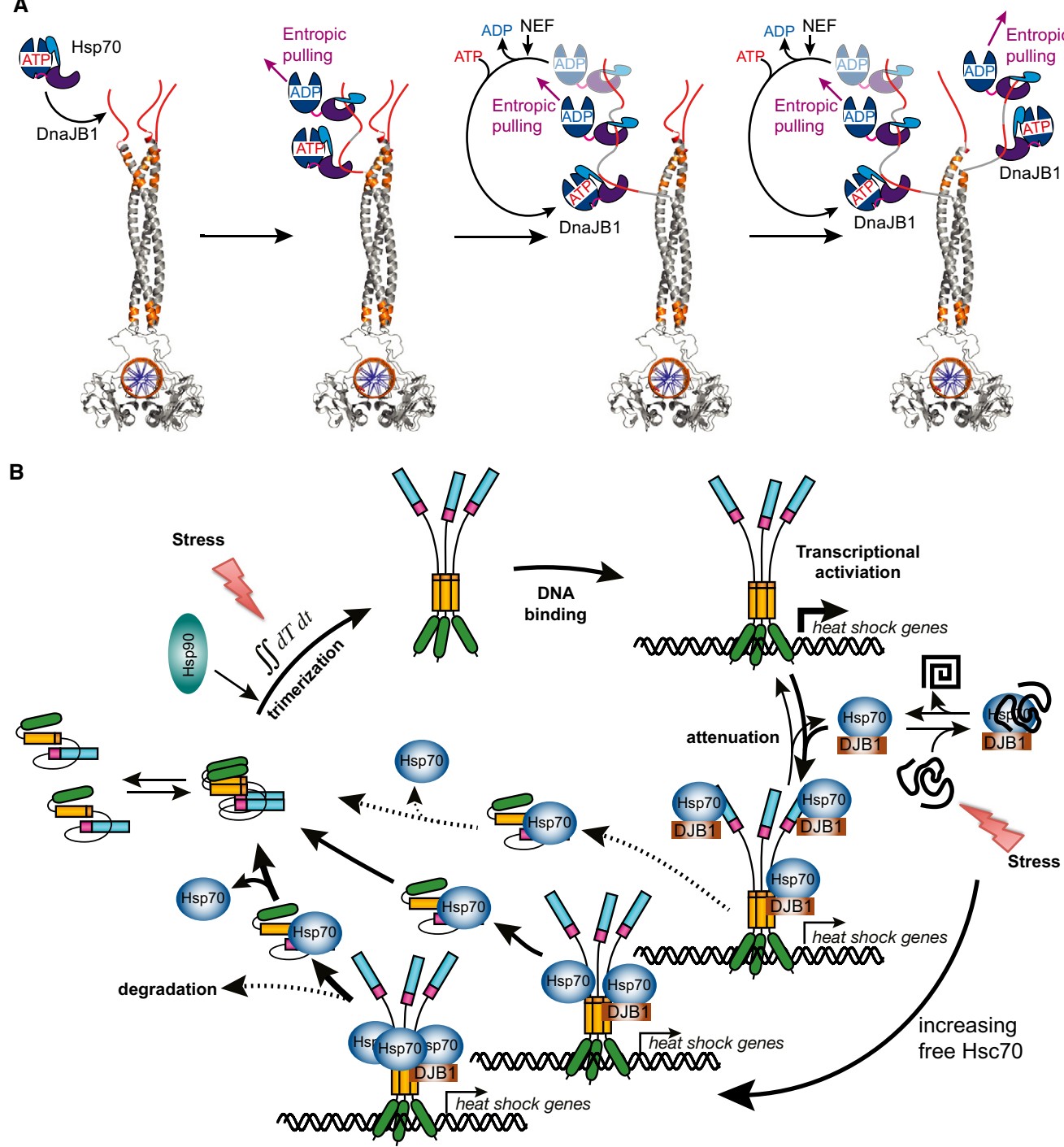

**Figure 8.**

**Figure 8. Model of the Hsp70/Hsc70-mediated regulation of the heat shock response.**

A Illustration of the stepwise disassembly of Hsf1 trimers by the entropic pulling action of Hsp70 and DnaJB1. Homology model of trimeric DNA-bound human Hsf1. Red, accessible Hsp70 binding sites; orange, Hsp70 binding sites not accessible in the Hsf1 trimer, only accessible after unfolding. Unzipping of the Hsf1 trimer may occur simultaneously on all three protomers.

B Hsf1 activation–attenuation cycle. In unstressed cells, Hsf1 is in monomer–dimer equilibrium, occasionally trimerizing depending on the local concentration. Trimeric Hsf1 either binds to HSE-promoter DNA driving heat shock gene transcription by releasing paused RNA polymerase or is disassembled immediately by Hsp70/DnaJB1 action (not shown for clarity). Hsp70 binds dynamically to the TAD of DNA-bound Hsf1 trimers attenuating transcriptional activity and binds to its HR-B proximal binding site, disassembling Hsf1 trimers and thus removing it from heat shock promoters. Release of Hsf1 from Hsp70 restarts this cycle. Upon heat shock Hsf1 trimerizes at elevated rates due to its thermosensory function, integrating over temperature and time at elevated temperatures. Simultaneously, Hsp70/DnaJB1-mediated attenuation and disassembly are slowed down, due to binding of Hsp70 to misfolded and aggregated proteins. Both parts of the cycle shift the Hsf1 pool rapidly to the DNA-bound active state, accelerating heat shock gene transcription. Elevating Hsp70 concentrations successively attenuate transcriptional activity of Hsf1 and disassemble Hsf1 trimers.

(Shi *et al*, 1998). Also, RNAi-mediated downregulation of DroJ1, the *Drosophila* homolog of DnaJB1, or Hsp70/Hsc70 in *Drosophila* SF2 cells induced heat shock gene transcription (Marchler & Wu, 2001). This core regulatory mechanism of the HSR is very likely conserved in all metazoa as suggested by a CLUSTAL Ω alignment of sequences from phyla across the metazoan clade (Fig EV5), showing that potential Hsp70 binding sites (segments with an Hsp70 score $\leq -5$) are found in HR-B and C-terminally adjacent to HR-B corresponding to the sites demonstrated here to be essential for monomerization of Hsf1. Interestingly, no potential Hsp70 binding site C-terminally adjacent to HR-B was found in *S. cerevisiae* and other *Ascomycota*, consistent with the view that yeast Hsf1 is constitutively trimeric and regulated in a different way by Hsp70, maybe only by binding to one or both TADs of yeast Hsf1. In contrast, in other fungi Hsp70 binding sites are predicted (Fig EV5).

Hsp90 and TRiC/CCT have also been suggested to regulate the HSR in mammalian cells (Zou *et al*, 1998; Neef *et al*, 2014). After Hsc70-mediated monomerization, Hsf1 could be transferred to Hsp90 or TRiC/CCT and thus involve these chaperones in the regulation of the heat shock response. However, in our *in vitro* DNA-binding assay with purified proteins we could not see any influence of Hsp90 on Hsf1 dissociation from DNA. We also had previously shown that Hsp90 does not prevent Hsf1 trimerization and DNA binding upon heat shock (Hentze *et al*, 2016). We cannot exclude that Hsp90 binds to Hsf1 while bound to HSE-DNA and prevents release of stalled RNA polymerase on heat shock genes.

The here proposed core cycle of Hsf1 activation and attenuation may be modulated at many sites, for example, by abundance regulation of Hsf1, by nuclear-cytoplasmic transport of Hsf1 and Hsc70 or nucleotide exchange factors like Apg2, and by posttranslational modifications in close proximity of the Hsc70 binding sites that may influence binding of Hsc70 to Hsf1. In particular, phosphorylation which was shown to accompany heat shock induction and to prolong heat shock gene transcription is expected to interfere with Hsc70 binding due to the fact that negative charges disfavor Hsp70 binding (Rüdiger *et al*, 1997). Interestingly, cancer genomics revealed mutations in Hsf1 that according to the Hsp70 binding score algorithm would be expected to lower the affinity of Hsc70 for Hsf1 (Appendix Fig S5). This prediction needs to be validated *in vitro* and *in vivo*, but if correct, these variant Hsf1 proteins should require higher Hsc70 concentrations for disassembly and attenuation of transcriptional activation. Hsf1 has been shown to promote tumorigenesis and cancer progression (Dai *et al*, 2007; Dai, 2018), and many tumor entities are addicted to high levels of molecular chaperones, like Hsp70, Hsp90, and Hsp27, and high levels of Hsp70 and Hsp90 have been associated with poor prognosis

(Calderwood & Gong, 2016; Yun *et al*, 2019). Thus, such amino acid replacements in Hsf1 could contribute to increased chaperone levels in cancer cells and thereby to cancer cell fitness and resistance to therapeutic intervention.

## Materials and Methods

### Protein expression and purification

Human Hsf1, Hsf1(1–408) (ΔTAD), Hsf1Δ(202–213), Hsf1-I202S,L203S,V205S, Hsf1-I209S,L211S,L213S, Hsf1_t10 (202–213 translocated by 10 residues toward the C-terminus), Hsf1_t20, Hsf1_t30, Hsf1_i201(DYKDDDDK), Hsf1_i211(DYKDDDDK), or Hsf1_i221 (DYKDDDDK) were purified as N-terminal His$_6$-SUMO fusions from *E. coli* BL21 (DE3) Rosetta cells after overproduction at 20°C for 2 h. Bacterial pellets were resuspended in lysis buffer (25 mM HEPES/KOH pH 7.5, 100 mM KCl, 5 mM MgCl$_2$ and 10% glycerol), drop-wise frozen in liquid nitrogen, disrupted using a Mixer Mill MM400 (Retsch), and resuspended in 200 ml Hsf1 lysis buffer supplemented with 3 mM β-mercaptoethanol, DNase, and protease inhibitors (10 µg/ml aprotinin, 10 µg/ml leupeptin, 8 µg/ml pepstatin, 1 mM PMSF). The resulting lysate was centrifuged (33,000× *g*, 4°C for 45 min) and the soluble fraction incubated for 25 min at 4°C with 0.4 g Protino Ni-IDA resin (Macherey-Nagel) in a rotation shaker. The resin was transferred to a gravity-flow column and washed with 100 ml of Hsf1 lysis buffer. Bound protein was eluted with Hsf1 lysis buffer containing 250 mM imidazole, 3 mM β-mercaptoethanol, and protease inhibitors. The His$_6$-SUMO tag was cleaved off by incubation with Ulp1 SUMO protease for 1 h at 4°C. The cleaved Hsf1 was further separated by size-exclusion on Superdex 200 HiLoad 16/60 column (GE Healthcare) and equilibrated with Hsf1 SEC buffer (25 mM HEPES/NaOH pH 7.5, 150 mM NaCl, 10% glycerol, 2 mM DTT). The fractions containing either monomeric or trimeric Hsf1 were concentrated to 10 µM, flash-frozen in liquid nitrogen, and stored at −80°C.

Human DnaJB1/Hdj1 and DNAJB1-H32Q,D34N were purified as a His$_6$-SUMO fusion from *E. coli* BL21 (DE3) Rosetta cells after overproduction for 3 h at 30°C. Cells were resuspended in DnaJB1 lysis buffer (50 mM HEPES/KOH pH 7.5, 750 mM KCl, 5 mM MgCl$_2$, 10% glycerol) supplemented with 3 mM β-mercaptoethanol, DNase, and protease inhibitors (10 µg/ml aprotinin, 10 µg/ml leupeptin, 8 µg/ml pepstatin, 1 mM PMSF). Bacteria were lysed using chilled microfluidizer at a pressure of 1,000 bar. The resulting lysate was centrifuged (33,000× *g*, 4°C for 45 min) and the supernatant incubated for 25 min at 4°C with 1.5 g Protino Ni-IDA resin

(Macherey-Nagel) in a rotation shaker. The resin was transferred to a gravity-flow column and washed with 200 ml of DnaJB1 lysis buffer and then with 100 ml of DnaJB1 SEC buffer (50 mM HEPES/KOH pH 7.5, 500 mM KCl, 5 mM $MgCl_2$, 10% glycerol). Protein was eluted with DnaJB1 SEC buffer containing 300 mM imidazole, 3 mM β-mercaptoethanol, and protease inhibitors. Fractions containing DnaJB1 were desalted (HiPrep 26/10 Desalting column, GE Healthcare) to SEC buffer and digested with Ulp1 protease overnight. After proteolytic cleavage, the protein was incubated with 1 g Protino Ni-IDA resin (Macherey-Nagel) for 25 min at 4°C to remove $His_6$-SUMO. The flow through was collected and further purified by size-exclusion on Superdex 200 HiLoad 16/60 column (GE Healthcare) equilibrated with DnaJB1 SEC buffer.

Human Hsc70 (HSPA8), Hsc70-K71M, Hsc70-V438F, and Hsp70 (HSPA1A) were purified as a $His_6$-SUMO fusion from overproducing *E. coli* BL21 (DE3) Rosetta. Cells were resuspended in Hsc70 lysis buffer (50 mM Tris pH 7.5, 300 mM NaCl, 5 mM $MgCl_2$, saccharose 10%) supplemented with 3 mM β-mercaptoethanol, DNase, and protease inhibitors (10 μg/ml aprotinin, 10 μg/ml leupeptin, 8 μg/ml pepstatin, 1 mM PMSF). Bacteria were lysed using a chilled microfluidizer at a pressure of 1,000 bar. The resulting lysate was centrifuged (33,000× *g*, 4°C for 45 min), and the supernatant was incubated for 25 min at 4°C with 2 g Protino Ni-IDA resin (Macherey-Nagel) in a rotation shaker. The resin was transferred to a gravity-flow column and washed with 200 ml of Hsc70 lysis buffer followed by high salt (Hsc70 lysis buffer but 1 M NaCl) and ATP (Hsc70 lysis buffer with 5 mM ATP) washes. Hsp70/Hsc70 were eluted with Hsc70 lysis buffer containing 300 mM imidazole, 3 mM β-mercaptoethanol, and protease inhibitors. Fractions containing Hsc70 were desalted (HiPrep 26/10 Desalting column, GE Healthcare) to HKM150 buffer (25 mM HEPES/KOH pH 7.6, 150 mM KCl, 5 mM $MgCl_2$) and digested with Ulp1 protease overnight. After proteolytic cleavage, the protein was incubated with 1.5 g Protino Ni-IDA resin (Macherey-Nagel) for 25 min at 4°C to remove $His_6$-SUMO. Flow through was collected, desalted to HKM150 buffer, concentrated to 50 μM, aliquoted, flash-frozen in liquid nitrogen, and stored at -80°C.

Human Hsp90α and Hsp90α-E47A were purified as a $His_6$-SUMO fusion from *E. coli* BL21 (DE3) Rosetta cells after overproduction overnight at 25°C. Cells were harvested by centrifugation (5,000× *g*, 4°C for 10 min), resuspended in Hsp90α lysis buffer (20 mM HEPES/KOH pH 7.5, 100 mM KCl, 5 mM $MgCl_2$, 10% glycerol) supplemented with 3 mM β-mercaptoethanol, DNase, and protease inhibitors (10 μg/ml aprotinin, 10 μg/ml leupeptin, 8 μg/ml pepstatin, 1 mM PMSF), and lysed using a chilled microfluidizer at a pressure of 1000 bar. The resulting lysate was centrifuged (33,000× *g*, 4°C for 45 min), and the supernatant was incubated for 25 min at 4°C with 1.5 g Protino Ni-IDA resin (Macherey-Nagel) in a rotation shaker. The resin was transferred to a gravity-flow column and washed with 200 ml of Hsp90α lysis buffer. Protein was eluted with Hsp90α lysis buffer containing 400 mM imidazole, 3 mM β-mercaptoethanol, and protease inhibitors. Fractions containing Hsp90α were subjected to Ulp1 cleavage and desalting during overnight dialysis to Hsp90α dialysis buffer (20 mM HEPES/KOH, pH 7.5, 20 mM KCl, 5 mM $MgCl_2$, 10% glycerol, 3 mM β-mercaptoethanol). After proteolytic cleavage, protein was incubated with 1.2 g Protino Ni-IDA resin (Macherey-Nagel) for 25 min at 4°C to remove $His_6$-SUMO. The flow through was subjected to anion

exchange chromatography (ResourceQ column, GE Healthcare, 20 mM HEPES/KOH pH 7.5, 5 mM $MgCl_2$, 10% glycerol, 20–1,000 mM KCl gradient in 16 CV) followed by size-exclusion on Superdex 200® HiLoad 16/60 column (GE Healthcare) equilibrated with Hsp90α SEC buffer (40 mM HEPES/KOH, pH 7.5, 50 mM KCl, 5 mM $MgCl_2$, 10% glycerol, 3 mM β-mercaptoethanol). The fractions containing Hsp90α were pooled, concentrated to 50 μM, aliquoted, flash-frozen in liquid nitrogen, and stored at -80°C.

Human Apg2 was purified as a $His_6$-SUMO fusion from *E. coli* BL21 (DE3) Rosetta cells after overproduction overnight at 30°C. Cells were subsequently harvested by centrifugation (5,000× *g*, 4°C for 10 min), resuspended in Apg2 lysis buffer (40 mM Tris/HCl pH 7.9, 100 mM KCl, 5 mM ATP) supplemented with 3 mM β-mercaptoethanol, DNase, and protease inhibitors (10 μg/ml aprotinin, 10 μg/ml leupeptin, 8 μg/ml pepstatin, 1 mM PMSF), and lysed using a chilled microfluidizer at a pressure of 1,000 bar. The lysate was centrifuged (33,000× *g*, 4°C for 45 min), and the supernatant was incubated for 25 min at 4°C with 2 g Protino Ni-IDA resin (Macherey-Nagel) in a rotation shaker. The resin was transferred to a gravity-flow column and washed with 200 ml of Apg2 lysis buffer. Protein was eluted with Apg2 lysis buffer containing 300 mM imidazole, 3 mM β-mercaptoethanol, and protease inhibitors. Fractions containing Apg2 were subjected to desalting to Apg2 desalting buffer (40 mM Tris pH 7.9, 100 mM KCl,). Desalted protein was supplemented with 10% glycerol, 5 mM ATP, and protein inhibitors, cleaved overnight with Ulp1 protease, and subsequently incubated with 1.5 g Protino Ni-IDA resin (Macherey-Nagel) for 25 min at 4°C to remove $His_6$-SUMO. The flow through was subjected to size-exclusion on Superdex 200 HiLoad 16/60 column (GE Healthcare) equilibrated with Apg2 SEC buffer (40 mM HEPES/KOH pH 7.6, 10 mM KCl, 5 mM $MgCl_2$, 10% glycerol) followed by IEC (ResourceQ column, 40 mM HEPES/KOH pH 7.6, 5 mM $MgCl_2$, 10% glycerol, 10–1,000 mM KCl gradient in 10 CV). The fractions containing Apg2 were pooled, supplemented with 5 mM ATP, aliquoted, flash-frozen in liquid nitrogen, and stored at −80°C.

## Preparation of MEF cell lysates

Confluent HSF1$^{-/-}$ mouse embryonic fibroblasts (McMillan *et al*, 1998) transfected with Hsf1wt or mutants expressing plasmids from 15-cm-diameter dishes (around 2*10$^6$ cells) were frozen at −80°C and subsequently resuspended in MEF's lysis buffer (20 mM HEPES/NaOH, pH 7.9, 25% glycerol, 0.42 M NaCl, 1.5 mM $MgCl_2$, 0.2 mM EDTA) freshly supplemented with protease inhibitors (10 μg/ml aprotinin, 10 μg/ml leupeptin, 8 μg/ml pepstatin, 0.5 mM PMSF) and 0.5 mM DTT. Cell suspensions were incubated for 45 min on ice and subsequently spun down at 100,000× *g*. Total protein concentration was estimated using Bradford assay. 30 μg of total protein was separated by SDS–PAGE, 15 μg were used for BN-PAGE, and 40 μg, where possible, were used for EMSA assays.

## Western blot

β-Actin was detected using anti-β-actin primary antisera (β-actin (D6A8) rabbit mAb, Cell Signaling Technology #8457, 1:2,000 dilution) and secondary fluorescently labeled antibodies (goat anti-rabbit IgG IRDye 680RD, Odyssay, 1:20,000 dilution).

Human Hsf1 was detected using anti-Hsf1 primary antisera (HSF1 (H-311) rabbit polyclonal IgG, Santa Cruz Biotechnology sc-9144, 1:1,000 dilution), secondary fluorescently labeled antisera (goat anti-rabbit IgG IRDye 680RD, Odyssay, 1:20,000 dilution), or HRP-conjugated antisera (peroxidase AffiniPure donkey anti-rabbit IgG, Jackson ImmunoResearch, 1:10,000).

In case of fluorescent antibodies, fluorescence was detected using the LICOR Odyssey Infrared Imaging System (700 nm channel) and quantified using Image Studio Lite Ver 5.2. In case of HRP-conjugated antibodies, chemiluminescence was detected using FUJI LAS-4000 fluorescence imager (Fuji Photo Film, Düsseldorf, Germany). Quantification was performed in ImageJ.

### Blue-native PAGE (BN-PAGE)

Protein samples pre-mixed with 4× BN-PAGE sample buffer (250 mM Tris/HCl, pH 6.8, 40% glycerol, 0.1% Coomassie Brilliant Blue G-250) were separated on 8% PAGE gels in Laemmli system (cold BN-PAGE running buffer: 25 mM Tris, 0.2 M glycine). Separation was carried out at constant current (20 mA per gel, 1 h, on ice). Gels were subsequently stained using staining solution (Quick Coomassie Stain, Serva) or used further for Western blot analysis: primary antibody (Hsf1 (H-311) rabbit polyclonal IgG, Santa Cruz Biotechnology, 1:1,000 dilution) and secondary fluorescently labeled antibody (goat anti-rabbit IgG IRDye 680RD, Odyssay, 1:20,000 dilution). Fluorescence was detected on LICOR Odyssey Infrared Imaging System (700 nm channel) and quantified using Image Studio Lite Ver 5.2.

### Hsf1 temperature response

Hsf1 (5 μM wild-type or mutant protein) was incubated at the given temperatures (4, 20, 25, 30, 35, 39, 42°C) for 10 min. Samples were subsequently analyzed using BN-PAGE. The obtained BN-PAGE gels were scanned and quantified using ImageJ. The data points were fitted to the thermal unfolding equation using GraphPad Prism software.

### EMSA assay

Hsf1 (100 nM Hsf1 trimer) was incubated with 50 nM double-stranded Cy3-labeled HSE-containing DNA, 5 μM Hsc70, 2.5 μM DnaJB1, 2 mM ATP, and 4 mM MgCl$_2$ in Hsf1 SEC buffer (25 mM HEPES/NaOH pH 7.5, 150 mM NaCl, 10% glycerol, 2 mM DTT). ATP/MgCl$_2$ mixture was added to the samples to start the reaction. Samples were subsequently mixed with glycerol (to 20% final concentration) and analyzed on pre-chilled and pre-equilibrated 1% agarose gel in 0.5xTBE buffer at 4°C (15 min 80 V and then 28 min 120 V). Labeled HSE-DNA was detected on a FUJI LAS-4000 fluorescence imager (Fuji Photo Film, Düsseldorf, Germany). In case of cell lysates, 80 nM double-stranded Cy3-labeled 3xHSEs were used. Samples were desalted to Hsf1-SEC buffer incubated for 10 min with DNA and subjected to separation and detection as above.

### Fluorescence spectroscopy

#### Binding of trimeric Hsf1 to heat shock elements (HSEs)

Fluorescently labeled double-stranded 3xHSEs were prepared by annealing of fluorescently labeled Alexa Fluor® 488-3xHSE sense oligonucleotides with 3xHSE antisense nucleotides (2 min at 70°C, 0.6°C/min stepwise decrease from 70 to 30°C). 5 nM 3xHSEs were titrated with trimeric Hsf1 at different concentration range (from 0.2 nM to 100 nM). Protein was mixed with 3xHSEs at 1:1 ratio on low volume 384-well plate (CORNING, REF3820). The plate was subsequently spun down at 1,000× $g$ for 1 min at RT. Fluorescence anisotropy of prepared samples was measured at 25°C using plate reader (CLARIOstar, BMG Labtech, Excitation, F:482-16, Emission, F:530-40). The data points were fitted to the quadratic solution of the law of mass action using GraphPad Prism software.

#### Hsf1 dissociation from heat shock elements (HSEs)

Fluorescently labeled double-stranded 3xHSEs were prepared by annealing of fluorescently labeled Alexa Fluor® 488-3xHSE sense oligonucleotides with 3xHSE antisense nucleotides (2 min at 70°C, 0.6°C/min stepwise decrease to 30°C). 5 μM Hsc70, 2.5 μM Hdj1/DnaJB1, 10 μM Hsp90α, 2 mM ATP, and 4 mM MgCl$_2$ in Hsf1 SEC buffer (25 mM HEPES/NaOH pH 7.5, 150 mM NaCl, 10% glycerol, 2 mM DTT) pre-incubated at 30°C for 30 min were mixed with 100 nM trimeric Hsf1 and 25 nM HSEs on low volume 384-well plate (CORNING, REF3820) in a final 20 μl reaction volume. The plate was subsequently spun down at 1,000× $g$ for 1 min at RT. Fluorescence anisotropy of prepared samples was measured at 25°C using a microplate reader (CLARIOstar, BMG Labtech, Excitation, F:482-16, Emission, F:530-40). The trimeric Hsf1 fraction of the gelfiltration was used for the experiments, if not stated otherwise in the figure legend. A single exponential equation with delay was fitted to the data using Prism (GraphPad software) (see Fig EV1E).

#### Hsf1 FLAG-variants dissociation from HSEs

Anti-FLAG antibody (SIGMA) was desalted to Hsf1 SEC buffer (25 mM HEPES/NaOH pH 7.5, 150 mM NaCl, 10% glycerol, 2 mM DTT) and incubated for 1 h at 25°C to reduce it. Reduced antibody was used in the dissociation experiment (described above) at 5 μM final concentration.

#### Hsc70 binding to fluorescently labeled peptides

Peptides were incubated with equimolar amounts of TCEP for 30 min at 25°C to reduce the C-terminal cysteine residue. 100 μM of peptide was mixed with 22-fold molar excess of fluorescein-5-maleimide (Thermo Fischer Scientific) in 25 mM HEPES/KOH pH 7.2 and incubated for 2 h at 25°C. Labeled peptides were separated from excess free dye on a Sephadex G-10 gravity column (GE Healthcare, 17-0010-01) equilibrated with HKM50 buffer (25 mM HEPES/KOH pH 7.6, 50 mM KCl, 5 mM MgCl$_2$). Labeled peptides at 1 μM final concentration were titrated with Hsc70 (0–60 μM final concentration) and incubated on low volume 384-well plate (CORNING, REF3820) in 20 μl final volume for 2 h at 25°C. Fluorescence anisotropy was measured using a CLARIOstar plate reader (BMG Labtech, Excitation, F:482-16, Emission, F:530-40). To avoid the formation of oligomeric species, Hsc70 was concentrated in the presence of 1 mM ATP. HKM150 (25 mM HEPES/KOH pH 7.6, 150 mM KCl, 5 mM MgCl$_2$) was used as assay buffer. The quadratic solution of the law of mass action was fitted to the data using GraphPad Prism.

#### Hsf1 trimers formation

Hsf1 can be purified as a trimer and monomer. Monomeric Hsf1 can be converted into trimeric Hsf1 by incubation at 42°C for 10 min. In

case of Hsf1 heterotrimer formation, monomers of Hsf1wt and Hsf1Δ(202–213) were pre-mixed at different ratios and subjected to a 10-min incubation at 42°C.

### Hydrogen exchange mass spectrometry—sample preparation

#### Effects of Hsc70 or Hdj1/DnaJB1 on Hsf1

6 μM Hsf1 (monomer or trimer) were pre-incubated for 30 min at 25°C in the presence of 12 μM Hsc70 or 9 μM DnaJB1 in Hsf1 SEC buffer (25 mM HEPES/NaOH pH 7.5, 150 mM NaCl, 10% glycerol, 2 mM DTT) and, subsequently, diluted 10-fold in $H_2O/D_2O$ buffer to a final volume of 100 μl and incubated for 30 s at 25°C. Deuteration reaction was quenched by adding 100 μl of ice-cold quench buffer (400 mM Na-phosphate, pH 2.2) and was immediately injected into the ice-cold HPLC system.

#### Effects of the Hsp70 system on Hsf1

10 μM Hsf1 trimers were pre-incubated in the presence of 40 μM Hsc70, 12 μM DnaJB1, 2 mM ATP, and 4 mM $MgCl_2$ in Hsf1 SEC buffer (25 mM HEPES/NaOH pH 7.5, 150 mM NaCl, 10% glycerol, 2 mM DTT) for 30 min at 25°C, subsequently, diluted 10-fold in $D_2O$ buffer to a final volume of 100 μl, and incubated for 300 s at 25°C. The deuteration reaction was quenched by adding 100 μl of the ice-cold quench buffer (2% formic acid) and was immediately injected into the ice-cold HPLC system.

### Performing hydrogen exchange mass spectrometry measurements

In the ice cooled HPLC set-up, the protein was digested online using a column with immobilized pepsin and the peptides were desalted on a C8 trap column (POROS 10 R2, Applied Biosystems, #1-1118-02) for 2 min and eluted over an analytical C8 column (Waters GmbH, 186002876) using a 10-min gradient from 5 to 55% acetonitrile. All experiments were performed using a maXis mass spectrometer (Bruker, Bremen, Germany) and analyzed with the Data Analysis software. The calculated centroid values were corrected for the back exchange using a 100% deuterated sample.

### Cell culture experiments

HSF1$^{-/-}$ MEF (mouse embryonic fibroblasts) were a kind gift of Prof. I.J. Benjamin (University of Utah, School of Medicine, Salt Lake City Utah, USA) (McMillan *et al*, 1998). The HeLa cells were a gift of Dr. K. Schönig (Weidenfeld *et al*, 2009). The cells were cultured in high-glucose Dulbecco's modified Eagle media (GIBCO, Thermo Fisher Scientific) with 10% fetal bovine serum (GIBCO, Thermo Fisher Scientific), 1% penicillin/streptomycin (Pen/Strep, Merck, Germany), and 1% minimum essential Eagle's medium (Merck, Germany) at 37°C and 5% $CO_2$. Stable and transient transfection was performed using TRansIT®-X2 Dynamic Delivery System (Mirus, Madison, USA) according to the manufactures instructions. HSF1$^{-/-}$ MEF P$_{HSPA6}$-mTagBFP and HeLa P$_{HSPA6}$-mTagBFP cells were constructed by transfection of HSF1$^{-/-}$ MEF and HeLa cells with pcDNA5-ble-P$_{HSPA6}$-mTagBFP, selection in DEMEM supplemented with Zeocin (InvivoGen 900 μg/ml) for 7–10 days and subsequently sorting by fluorescence-activated cell

sorting (FACS) on a BD FACS Aria III (BD Biosciences, excitation 407 nm, emission filter 450/40 nm) for BFP-positive single cells. Individual clones were tested for successful integration of the Hsf1 activity reporter by transient transfection with pIRES-HSF1_hrGFP.

To test Hsf1 activity of Hsf1 wild-type and mutant proteins, HSF1$^{-/-}$ MEF P$_{HSPA6}$-mTagBFP and HeLa P$_{HSPA6}$-mTagBFP cells were transiently transfected using Roti®FectPlus (Carl Roth GmbH, Karlsruhe, Germany) or electroporation with plasmids pIRES_hrGFP, pIRES-luciferase_hrGFP; pIRES-HSF1_hrGFP, pIRES-HSF1Δ(202–213)_hrGFP, pIRES-HSF1-I202S,L203S,V205S_hrGFP, pIRES-HSF1-I209S, L211S,L213S_hrGFP, pIRES-HSF1-L465S,V466S,Y468S_hrGFP, pIRES-HSF1-L473S,F474S,L475S,L476S_hrGFP, pIRES-HSF1Δ(425-438)_hrGFP, and pIRES-HSF1-M427S,L249S,L432S,L436S_hrGFP, and grown for 24 h at 37°C, and cells were split into two 10-cm dishes and grown for additional 24 h at 37°C. One of the two dishes of each transfection was treated at 43°C for 1 h followed by 3 h at 37°C, and subsequently, BFP fluorescence and GFP fluorescence were analyzed by flow cytometry (BD FACS Canto; excitation 405 and 488 nm; emission filter 450/50 and 530/30 nm).

### Quantification and statistical analysis

All biochemical assays were performed at least three times independently. Data were analyzed with GraphPad Prism 6.0 (GraphPad Software). Statistical significance was estimated by ANOVA or *t*-tests as indicated in Figure legends. For quantification, ImageJ or Image Studio Lite Ver 5.2 was applied.

## Data and software availability

- HX-MS data: PRIDE PXD017406 (https://www.ebi.ac.uk/pride/archive/projects/PXD017406)

- Images shown in Figs 2B and 5D, and EV1C and EV3A, C, E and G, Appendix Figs S3 and S4: Mendeley https://doi.org/10.17632/jbbs52642f.1 (https://data.mendeley.com/datasets/jbbs52642f/draft?a=710f2c9e-1cc7-491f-a7ed-7139ea90f42d)

- Flow cytometry data: FlowRepository (http://flowrepository.org/id/RvFr4XRnZUs8vZKtJDITOczC5ghFZ3BuWwqiQS4i0VsFYvvfwj6uWJeAt44a5jhd.
  http://flowrepository.org/id/RvFrTKqZqYoUq1oSLxBYFfo28Ae4Q8RatrUdK5SS2hLx5Hd13LsSE7z93VieFxCJ.
  http://flowrepository.org/id/RvFrx1YEgg3Faf1LyjkEln6VJkpDy7gBytgbEjcTHQM9DSqDjRiLeWIC9GXyKIGR.
  http://flowrepository.org/id/RvFrp4RNI2iCWIpNPKVd307q12LQlEAVU8gOqaCkleY1y61l1uWv9aUzhMrdUtuv.
  http://flowrepository.org/id/RvFrI1qVmaaWSkzhpHnjRjIv8dIrwwme3ebFgxqQk4LklPUpSHycvYKvHbFxT3pAw.
  http://flowrepository.org/id/RvFrYTFaUfYIjftam3SAMBASsIJojQ9xVwClLQCsLeqxHzcWy50RiGUzInKIslel.
  http://flowrepository.org/id/RvFrjGXlOnBz9lfowJLwBAxO6y5ERRnPFlXlkhalztTZFPbfl0gAH7IEVMOcCldY.
  http://flowrepository.org/id/RvFrrrusbW5XmcXzwMEwUhXUqRnr99Y10pfZP5bzCnVnla4MYoWTLJfMjMksGags.

Expanded View for this article is available online.

## Acknowledgements

We thank Dr. B. Bukau and Dr. S. Bergink for helpful discussions. We thank Dr.
G. Stöcklin for providing the HSF1$^{-/-}$ MEFs, Dr. M. Langlotz for help in the
FACS facility, Dr. T. Ruppert and N. Lübbehusen for help in the core facility for
mass spectrometry and proteomics, and S. Hennes and A. Müller for excellent
technical assistance. This work was funded by the Deutsche Forschungsge-
meinschaft (SFB 1036 TP09).

## Author contributions

Conceptualization, MPM; Methodology, LLB, SWK, and MPM; Investigation,
LLB, SWK, and MPM; Writing—Original Draft, SWK and MPM; Writing—Review
& Editing, MPM, SWK, and LLB; Funding Acquisition, MPM; and
Supervision, MPM.

## Conflict of interest

The authors declare that they have no conflict of interest.

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
