## [Review Process File · The EMBO Journal]

Feedback regulation of heat shock factor 1 (Hsf1) activity by Hsp70-mediated trimer unzipping and dissociation from DNA

Szymon Kmiecik, Laura Le Breton, and Matthias Mayer
DOI: 10.15252/embj.2019104096

Corresponding author(s): Matthias Mayer (M.Mayer@zmbh.uni-heidelberg.de)

Review Timeline:

Submission Date:	23rd Nov 19
Editorial Decision:	17th Dec 19
Revision Received:	17th Mar 20
Editorial Decision:	9th Apr 20
Revision Received:	19th Apr 20
Accepted:	24th Apr 20

Editor: Stefanie Boehm

Transaction Report:

Prof. Matthias P. Mayer
University of Heidelberg
ZMBH
Im Neuenheimer Feld 282
Heidelberg, Baden-Wuerttemberg 69120
Germany

17th Dec 2019

Re: EMBOJ-2019-104096
Molecular mechanism of attenuation of heat shock transcription factor 1 activity

Dear Prof. Mayer,

Thank you for submitting your manuscript on the function of Hsc70 in attenuation of Hsf1 for consideration by The EMBO Journal. We have now received three referee reports on your study, which are included below for your information.

As you will see, the reviewers are overall positive and acknowledge the study's contribution to the field and its quality. Nonetheless they also raise some concerns that would need to be either addressed experimentally or discussed in a revised manuscript. In particular, the referees find that it would be important to provide a more detailed description of the experimental approaches and the conclusions drawn from them, in particular with respect to Figure 4 and 5. In addition, referee #2 (point 5) and #3 (point 5) have raised two issues regarding the MEF-based experiments that should be addressed. To further strengthen the results in a cellular model, experiments that demonstrate the presence oligomeric forms (2-5) and that assess the response and recovery upon heat shock should be included. Should you be able to adequately address these issues, as well as responding to the more specific concerns raised by each of the referees, we would be happy to consider this study further for publication. Therefore I would like to invite you to prepare and submit a revised manuscript. Please note that it is our policy to allow only a single round of major revision and that it is therefore important to clarify all key concerns raised at this stage.

Please feel free to contact me should you have any further questions regarding the revision. Thank you for the opportunity to consider your work for publication. I look forward to receiving your revised manuscript.

Kind regards,

Stefanie Boehm

Stefanie Boehm
Editor
The EMBO Journal

Referee #1:

Kmiecik et al show, in a very impressive and detailed manner, how the activity of the heat shock factor-1 (HSF1) is attenuated by the Hsp70 chaperone machine. Whereas the regulation of HSF-1 has been studied for a few decades already and whereas it has been repeatedly suggested that Hsc70 plays a role in the attenuation of HSF-1 activity, no direct proof for, let alone insights in the mechanism of, such regulation has been provided up to date.

In this MS, it is shown how trimeric HSF-1 is monomerized by successive binding of Hsc70 to multiple site in HSF-1. Moreover, data are provided where in HSF-1 Hsc70 can bind, what the affinities are, and how the un-pulling/unzipping of HSF-1 trimers depends on the obligate Hsc70 co-factors (JDs and NEFs).

Mechanistically, the authors show that Hsc70-mediated unzipping is due entropic pulling by cycles of binding/release of Hsc70 to HSF-1. Hereby, the data also for the first time provide evidence for and insight in the model of entropic pulling, a mechanism that also has been proposed for Hsp70 actions during protein translocation and in protein disaggregation.

Finally, the authors show, using HSF-1 knockout lines, that this mode of action of the Hsp70 chaperone machine is also important for activation/attenuation of HSF-1 in mammalian cells.

The data are nicely presented and largely explained in an excellent manner and also resolved some conflicting views on HSF-1 regulation (e.g. role of Hsp90). The final figure 7 represents an elegant integration of the data presented in this MS, combined with the existing literature data, thus providing a nice, updated and elegant model on the (auto)regulation of the heat shock response.

Detailed comments:

Major:

- The authors show that the reactions on HSF-1 regulation are dependent on DNAJB1 and Apg2. Whereas this is correct, I may be interpreted as if these family members in the much larger families of J-domain proteins and NEFs are the only and thus required members for this reaction. Whereas I think it goes beyond the scope of this paper to test other J-domain proteins or NEFs, I feel the authors should make a disclaimer on this point.
- Page 12, figure 4. Here it is stated that all sites protected by Hsc70 are also protected by DNAJB1 and consistent with the idea that J-domain proteins bind the client first. Whereas this is indeed the current model, I first of all do not see how the authors deduce it from the data in figure 4, where no data on DNAJB1 alone are provided. Moreover, the predicted binding profiles for J-domain proteins (DNAJB1) and Hsc70 seem not to be identical and would not only suggest that DNAJB1 likely would bind to more sites on HSF-1 but also, at slightly different position in the protein, maybe adjacent to the Hsc70 binding site. Is this indeed what the authors observe and agree with?
- Page 13, Figure 4: I really find the data on HSF-1 mutants affecting Hsc70 mediated dissociation truly impressive and revealing.

Minor:

- Page 7: The statement that HSF-1 dissociation rates from DNA (their in vitro data) occurs with similar kinetics as attenuation of the HSR in HeLa cells (Abravaya et al 1991) seems a bit an overstatement as these kinetics (at least in cells) are highly dependent on the severity of the heat shock.

- Page 11, figure 3: Whereas the data clearly show lack of Hsc70 binding to amino acids 359-439, one still wonders why this has been a very prominent Hsc70 binding site in previous studies (Shi et al 1998). Can the authors explain this? Is there a methodological issue underlying this difference?
- Page 14, FigS4A: the sentence 'We simulated this situation and found the corresponding curves do not fit our experimental data' is a bit confusing. I suggest revising as follows: 'However, using this model, the simulated curves did not fit our experimental data'.
- Page 17: the sentence "Nevertheless, Hsc70-mediated.....than for Hsfwt" lack a reference to the corresponding data (Figure 5G).

Referee #2:

Summary:

HSF1 activation involves two steps; transition of a monomer to a DNA-binding trimer and acquisition of transcriptional activity. These steps are negatively regulated by chaperones including HSP70 and HSP90, but detailed mechanisms are unclear yet. In this manuscript, Mayer and colleague purified trimeric HSF1 and found using fluorescence polarization assay with Alexa488-labeled DNA that HSF1 trimer bound to DNA was dissociated from it by HSC70. Simultaneously, HSF1 trimer was monomerized. Authors then identified five binding sites of HSC70 within HSF1, which included HR-B proximal site and the site in activation domain, using hydrogen exchange mass spectrometry. Furthermore, three segments encompassing the trimerization domain was deprotected in the presence of HSC70, suggesting that the helices were unfolded. Authors determined that the HR-B proximal site (residues 202-213) was required for HSF1 monomerization by HSC70. Because the trimerization domain contained several potential HSC70 binding sites, which promoted HSF1 monomerization, they proposed that HSC70 monomerizes HSF1 trimer by entropic pulling unzipping of the trimerization domain. Reporter analysis using HSPA6 promoter-driven gene in cells showed that the HR-B proximal site is important for attenuation of the reporter gene transcription. Thus, authors proposed a model of HSC70-mediated monomerization of HSF1 trimer bound to DNA.

Unexpectedly, this study first shows that HSC70 monomerizes HSF1 and dissociates HSF1 from DNA by binding to it directly, and proposes the concept of entropic pulling for HSC70 action on the HSF1 trimerization domain, based on well-controlled and skillful experiments. Their observations greatly extend our understanding of regulation of HSF1 activity and contribute a lot to the field. I have only some concerns to be addressed as below.

Major comments:

1) Authors performed unique in vitro polarization DNA-binding assay, and first showed that HSC70 dissociated HSF1 from DNA (Fig. 1D). Could the same result be observed using Electrophoresis Mobility Shift Assay? Or, they detected HSC70-mediated dissociation because of this unique assay? They should at least comment on it in the text (P7, second paragraph).

2) Authors state that "Most of Hsf1 exhibited an electrophoretic mobility that is in between the trimeric and the monomeric state, presumably bound to Hsc70 (P9, line 10)." Are these bands contains HSC70? Authors should examine the presence of HSC70 in these bands, for example, by western blotting using HSC70 antibody or by checking retardation of them after addition of HSC70 antibody. In addition, did HSF1 dimer appear in the presence of HSC70 as they proposed transient

dimerization of HSF1 previously (Hentze et al, 2016)?

3) Authors state "To substantiate this hypothesis, we inserted a FLAG epitope between HR-B and the Hsc70 binding site or at 10 and 20 residues distance to HR-B (P16, line 8)." What do this experiment mean? Why Did FLAG antibody disturb HSC70 binding to the HSC70-binding site? Authors should explain it in more detail.

4) Authors state the results shown in Fig. 5 (P17). However, it is difficult to understand it because they did not indicate each HSF1 mutant used in Fig. 5C. For example, reporter activity in the presence of HSF1-202SSS is similar to that in the presence of WT, whereas reporter activity in the presence of HSF1-209SSS is higher. How can we interpret this result? Which HSF1 mutant represents one lacking HSC70-binding site in the TAD among 4 mutants? Please describe the results more carefully.

5) In Fig. 5, authors should show oligomeric form of at least some HSF1 mutants in vivo in cells. The results of expression level and in vivo oligomeric state should strength their proposal.

Minor comments:

1) Authors state "These data suggest that binding of Hsc70 to this region contributes to Hsf1 monomerization (P17, line 18)." Please indicate clearly "this region". It is located within the HR-B.

2) Fig. 5G should be referred in P17, line 18.

3) In Keywords section, it would be better to include Hsc70. Also, DnaJB1/Hdj1 could be replaced with DnaJB1/Hsp40 or DnaJB1/Hdj1/Hsp40.

4) It would be better to reconsider title to be more specific. For example, HSC70 dissociates HSF1 trimer from DNA by unzipping the trimerization domain.

Referee #3:

Heat shock transcription factor HSF1 is the master regulator of gene expression under a great variety of stress conditions, and recent genome-wide studies on different model systems have revealed its versatile targets in cell stress, growth, development, and disease, thereby extending its physiological impact far beyond the heat shock response or proteotoxic stress response. Yet, many fundamental questions of the activation and de-activation mechanisms of this important transcriptional regulator have remained unanswered in a rigorous manner. In this manuscript, the authors systematically go through past models of the attenuation phase of HSF1 DNA-binding and what evidence the models were based on. The authors then provide with their convincing data on which a new model is based on, while explaining how previous data still fits into their new model. The manuscript presents the minimum components required for HSF1 dissociation from the DNA, and the obtained results will enable a multitude of forthcoming studies to elaborate how these components are regulated in cells to attenuate the heat shock response. In fact, the results in this manuscript reveal that Hsc70-mediated dissociation makes the biggest contribution to attenuation of heat shock gene transcription in a cell culture model. Furthermore, the authors state that it might be the first paper with rigorous tests of the entropic pulling hypothesis mode of Hsp70 action and that their model explains results from many studies from the past 40 years. The impact of this

study is certainly high, but the authors could make it easier for the reader to understand how important the results are. Now, the importance is mentioned quite briefly in the Discussion only.

The manuscript is mostly well written, but the "modest" Abstract and the end of Introduction do not make justice for the content and impact of the study. Since the methodology of the study is the key for mechanistic understanding of the HSF1 attenuation mechanism, the authors could explain the basis of the fluorescence polarization assay in a more detail.

Specific comments:

1. The figure legend of Figure 2 is hard to follow, especially for panel C.
2. In Figure 4A, it is possible to distinguish five or six peaks with value below -5. On page 11, it is stated that "...two of the segments (200-213 and 442-474) protected from hydrogen exchange by Hsc70 covered sequences that fitted properties of strong Hsp70 binding sites...", which is clear but why are the other peaks at -5 not addressed?
3. In Figure 4B, four peptides display $\gg 60 \mu\text{M}$. Does this refer to the peptides described in the text "...To the other protected segments Hsc70 did not have any measurable affinity..."? If this is the case then why does the text refer to "For all four potential binding sites we could determine a KD between ...", when the figure displays five peptides that have a measurable Kd. Please, provide a more clear description of these results.
4. In the Results section "The number of available Hsc70 binding sites determines the rate of Hsf1 monomerization" (p. 13-14), the description of experimental set-ups and results is difficult to understand. This part of the text needs improving considering the importance of the obtained results.
5. Although the experiments are mainly performed in vitro, there is one critical experiment linking the results to a cell-based model system. As shown in Figure 6, HSF1^{-/-} MEFs were stably transfected with a heat shock reporter and subsequently they were transiently transfected to express either wild-type HSF1 or HSF1 where the HR-B (within the trimerization domain of HSF1) proximal Hsc70-binding site or the TAD (transactivation domain of HSF1) distal Hsc70-binding site was disrupted. Based on the results from the reporter assay, the authors conclude that binding of Hsc70 to the HR-B proximal Hsc70-binding site is important for attenuation of heat shock gene transcription, whereas they suggest that the Hsc70-binding site in the TAD contributes to attenuation indirectly, presumably through interference of HSF1 interaction with the core transcription machinery. This conclusion is intriguing and raises an important question why the experiment shown here was performed in unstressed cells only. It would be important to perform a corresponding experiment in cells exposed to heat shock and followed by a recovery. This experiment would certainly provide more rigor to the proposed model of attenuation.
6. In the end of the Discussion, it is postulated that "In particular, phosphorylation which was shown to accompany heat shock induction and to prolong heat shock gene transcription is expected to interfere with Hsc70 binding due to the fact that negative charges disfavor Hsp70 binding (Rudiger et al. 1997)". However, in the study by Budzyński et al. 2015 (doi: 10.1128/MCB.00816-14), it was shown that phosphorylation within the so-called regulatory domain (RD) is not required for HSF1 activation. It would be interesting to know how the authors think of a possibility that a lack of phosphorylation is able to promote or facilitate HSF1 activation, keeping in mind that less phosphorylation should in this case attract more Hsc70 binding.

Hsp70s dissociate Hsf1 trimers from DNA by unzipping the trimerization domain

Szymon W. Kmiecik, Laura Le Breton, and Matthias P. Mayer

Answers to the referees' comments

Referee #1:

Kmiecik et al show, in a very impressive and detailed manner, how the activity of the heat shock factor-1 (HSF1) is attenuated by the Hsp70 chaperone machine. Whereas the regulation of HSF-1 has been studied for a few decades already and whereas it has been repeatedly suggested that Hsc70 plays a role in the attenuation of HSF-1 activity, no direct proof for, let alone insights in the mechanism of, such regulation has been provided up to date.

In this MS, it is shown how trimeric HSF-1 is monomerized by successive binding of Hsc70 to multiple site in HSF-1. Moreover, data are provided where in HSF-1 Hsc70 can bind, what the affinities are, and how the un-pulling/unzipping of HSF-1 trimers depends on the obligate Hsc70 co-factors (JDs and NEFs).

Mechanistically, the authors show that Hsc70-mediated unzipping is due entropic pulling by cycles of binding/release of Hsc70 to HSF-1. Hereby, the data also for the first time provide evidence for and insight in the model of entropic pulling, a mechanism that also has been proposed for Hsp70 actions during protein translocation and in protein disaggregation.

Finally, the authors show, using HSF-1 knockout lines, that this mode of action of the Hsp70 chaperone machine is also important for activation/attenuation of HSF-1 in mammalian cells.

The data are nicely presented and largely explained in an excellent manner and also resolved some conflicting views on HSF-1 regulation (e.g. role of Hsp90). The final figure 7 represents an elegant integration of the data presented in this MS, combined with the existing literature data, thus providing a nice, updated and elegant model on the (auto)regulation of the heat shock response.

We thank the referee for this positive evaluation of our work.

Detailed comments:

Major:

- The authors show that the reactions on HSF-1 regulation are dependent on DnaJB1 and Apg2. Whereas this is correct, I may be interpreted as if these family members in the much larger families of J-domain proteins and NEFs are the only and thus required members for this reaction. Whereas I think it goes beyond the scope of this paper to test other J-domain proteins or NEFs, I feel the authors should make a disclaimer on this point.*

The referee is completely correct. We do not want to claim that other members of the J-domain protein family or NEF-families could not provide similar action on Hsf1 regulation or might even be better in monomerization. We chose DnaJB1 because it is one of the heat-

induced J-domain proteins, and Apg2 is most frequently used in disaggregation assays. Investigations of the influence of other J-domain proteins and nucleotide exchange factors on Hsc70-mediated Hsf1 dissociation from DNA and monomerization will be part of a future study. We have made the disclaimer in the manuscript in the discussion section:

“Of note, we do not exclude that J-domain proteins other than DnaJB1 or nucleotide exchange factors other than Apg2 could also support Hsc70-mediated monomerization of Hsf1.”

• *Page 12, figure 4. Here it is stated that all sites protected by Hsc70 are also protected by DNAJB1 and consistent with the idea that J-domain proteins bind the client first. Whereas this is indeed the current model, I first of all do not see how the authors deduce it from the data in figure 4, where no data on DNAJB1 alone are provided.*

We apologize for the misunderstanding. We drew the conclusion not from figure 4 but from figure 3A right bar graph, where HX-MS data for the interaction of DnaJB1 with Hsf1 are provided. In the original manuscript we inadvertently omitted the reference to figure 3. This reference has now been included. We also moved this statement to an earlier paragraph to avoid the misunderstanding.

Moreover, the predicted binding profiles for J-domain proteins (DNAJB1) and Hsc70 seem not to be identical and would not only suggest that DNAJB1 likely would bind to more sites on HSF-1 but also, at slightly different position in the protein, maybe adjacent to the Hsc70 binding site. Is this indeed what the authors observe and agree with?

The referee is correct. Peptide library scanning revealed that the recognition motif for the *E. coli* J-domain protein DnaJ deviates somewhat from the recognition motif for *E. coli* DnaK, being slightly longer (8 residues instead of the 5 residues core), more preference for aromatic residues and without strong bias against negatively charged residues, which contrasts the strong bias against negatively charged residues of Hsp70s (Rüdiger *et al*, 2001). However, 80% of the found DnaJ binding sites were also DnaK binding sites. A recent publication of the structure of bound PhoA and MBP on J-domain proteins (*T. thermophilus* DnaJ, yeast Ydj1 and yeast Sis1 the orthologue of DnaJB1) are completely consistent with the earlier findings (Jiang *et al*, 2019). In this publication several binding sites on the J-domain protein were identified showing that J-domain proteins bind through avidity to protein substrates, consistent with earlier observations that J-domain proteins bind proteins with higher affinity than peptides. Interestingly, the protected segments in the HX-MS experiments contain two potential binding sites as mentioned in the manuscript. Thus, it is conceivable that DnaJB1 binds both binding sites and then one is transferred to Hsc70. Since this is highly speculative and we do not have data for such a mechanism and due to space limitations, we did not include this speculation in the manuscript.

• *Page 13, Figure 4: I really find the data on HSF-1 mutants affecting Hsc70 mediated dissociation truly impressive and revealing.*

We thank the reviewer for this assessment of our work.

Minor:

• *Page 7: The statement that HSF-1 dissociation rates from DNA (their in vitro data) occurs with similar kinetics as attenuation of the HSR in HeLa cells (Abravaya et al 1991) seems a*

bit an overstatement as these kinetics (at least in cells) are highly dependent on the severity of the heat shock.

With our statement “The kinetics of Hsc70-mediated Hsf1 dissociation from DNA were very similar to the kinetics with which Hsf1-mediated transcription activation and DNA binding of Hsf1 decreased in HeLa cells during recovery after a heat shock” we did not want to claim that recovery from a heat shock *in vivo* and Hsp70-mediated dissociation of trimeric Hsf1 *in vitro* with purified proteins are completely comparable, since at the end of a heat shock, depending on the actual temperature at which the heat shock was performed, there might still be some misfolded proteins and protein aggregates that need to be cleared away before the Hsp70 machinery is able to take Hsf1 off the DNA. We just wanted to express that there are not orders of magnitude differences in kinetics. Very often *in vitro* kinetics are slower than *in vivo* kinetics due to multiple factors as for example reduced protein activity *in vitro* due to purification related artefacts, suboptimal buffer conditions, lack of posttranslational modifications of recombinant proteins purified from *E. coli*, missing activating factors, missing other regulatory factors, missing molecular crowding. We were actually struck by the similarity in time scale of Figure 5 in Abravaya et al. 1991 and our Figure 1d. The half-life of bound Hsf1 was estimated to be 10 to 15 min during recovery and the half-life of Hsf1 dissociation in our *in vitro* experiments was 17 min (Fig. 1d; $T_{1/2} = 8$ min of exponential phase + 9 min delay). Of note, our experiments were performed at 25°C not at 37°C as the experiments in cell culture and for technical reasons our microtiter plates are not immediately at 25°C when the measurement in the plate reader is started as there is no water heating of the plate but air heating, and a delay of 5 to 10 min is unavoidable.

In the revised manuscript we replace “with very similar kinetics” with “on the same time scale as”. This should be taken as a statement of the fact that our *in vitro* experiments are not orders of magnitude slower than *in vivo*.

• Page 11, figure 3: *Whereas the data clearly show lack of Hsc70 binding to amino acids 359-439, one still wonders why this has been a very prominent Hsc70 binding site in previous studies (Shi et al 1998). Can the authors explain this? Is there a methodological issue underlying this difference?*

The short answer is that we cannot explain this apparent discrepancy. To go into a bit more detail of our view, we first need to recapitulate the evidence of Shi et al.. Based on an earlier observation that transcriptional activity seems to decline faster during attenuation than DNA binding of Hsf1 (Kline & Morimoto 1997), Shi et al assumed that there must be something binding to the transactivation domain to inhibit activation of transcription prior to dissociation of Hsf1 from DNA. To identify such a factor, they performed pull-down assays with the region 395-503 fused to GST, assuming that the transactivation domain starts at 395. With this fusion construct they coprecipitated Hsp70 and such coprecipitation was slightly diminished when 395 to 411 or 395 to 425 was deleted and completely lost when region 395 to 439 was deleted from this construct, leading Shi et al to the conclusion that the Hsc70 binding site must be somewhere between 425 and 439 (see the reproduced Fig.4 from Shi et al. 1998).

Figure 4. Mapping the Hsp70 binding site within the HSF1 activation domain. (A) Schematic diagram of wild-type and deletion mutants of HSF1 activation domain. Constructs I (wild type), and II–V (deletion mutants) are indicated as the regions of HSF1 activation domain fused to GST. The boundaries of each construct and the levels of Hsp70-binding activity are indicated. (B) A potential Hsp70 binding site is located from amino acid residue 425 to 439 of the HSF1 activation domain. HSF1–GST fusion proteins were incubated with HeLa whole cell extracts (WCE). The presence of Hsp70 as bound protein was detected by Western blot analysis with Hsp70-specific antibody 3A3. Purified recombinant Hsp70 and an aliquot of HeLa whole cell extracts were included as positive controls. (C) SDS-PAGE of each purified GST fusion protein visualized by Coomassie blue staining. Molecular weight markers (MW) are included.

From these data we concluded as did Shi et al that the binding site must be between 425 and 439 as upon deletion of this region the interaction is lost completely. However, our experiments in cell culture demonstrate that neither deleting the region 425–439 nor replacing the hydrophobic residues in this region with serine increases transcriptional activity of Hsf1 as compared to wild type Hsf1, indicating that this region is not important for transcriptional attenuation.

Following the remark of this reviewer we reinvestigated all the evidence. The slight reduction of interaction between Hsp70 and Gst-Hsf1(411–503) as compared to Gst-Hsf1(395–503) could suggest that there may be a second binding site in this region. The prediction algorithm predicts a reasonable binding site between residues 410 and 423 (see our manuscript Fig. 4; sequence DTSALLDLFSPSV with a score of -8.35). Our HX-MS data contain two peptic peptides covering this region, amino acids 403–414 and 415–429 (see Fig. 3A), indicating that pepsin cleaved exactly between L414 and L415. After carefully studying crystal and NMR structures of the substrate binding domain of Hsp70s in complex with a peptide substrate or the unfolded C-terminus of the construct (Zhu *et al*, 1996; Morshausen *et al*, 1999) we realized that only a single amide proton within the bound peptide backbone segment is protected from a hydrogen bond and this amid proton is N-terminal of the bound leucine. If therefore leucine 15 within the 13mer peptide DTSALLDLFSPSV would be bound by Hsc70

then the protected proton would remain undetected as any incorporated deuterium in this position in Hsf1 in the absence of Hsc70 would be lost during cleavage with pepsin. We therefore added a disclaimer to our manuscript:

“Of note, protection in HX-MS experiments do not necessarily mean direct binding. Protection could also be caused allosterically by changing the conformation of the client protein. Also, we may miss actual binding sites, if the region is not covered by the peptic peptides or if pepsin cleaves within the bound segment leading to a fast back-exchanging amino group. In addition, the protected segments were larger than the segment bound by the substrate binding domain of an Hsp70, introducing an uncertainty as to where exactly Hsc70 was bound within the protected segment.”

However, even if the segment between 395 and 411 contained a potential Hsc70 binding site, this does not diminish our conclusions that no site in the transactivation domain is important for Hsc70-mediated Hsf1 monomerization, since a C-terminal truncation construct, Hsf1(1-408) that had the potential Hsc70 binding site between 410 and 423 and our newly discovered binding sites from amino acids 461 to 471 and from 468 to 480 deleted, was still monomerized with similar kinetics as Hsf1wt.

Why did Shi et al. not detect the binding sites we found between 461 and 480? The reason might be the use of Gst-fusion constructs. Such an approach, although quite common in cell biology, does not take into account the structure of the protein and potential interactions of this region with other parts of the protein, not included in the fusion construct. In fact, amino acids 395 to at least 409 are part of the heptad-repeat HR-C that represses trimerization. When we deleted the region 409 to the end of Hsf1, we noticed that this protein is constitutively trimeric and no monomeric version of this truncation construct could be purified. Nevertheless, Hsf1(1-408) was monomerized by Hsc70, DnaJB1 and ATP as shown in Fig 4C of our revised manuscript. These observations suggest that this region is important for interaction with HR-A&B at least in the monomeric state. Our HX-MS data suggest that the backbone amid protons in the region between 395 and 439 are largely accessible in the trimeric state. However, we cannot exclude that there are sidechain interactions between the transactivation domain and for example the regulatory region or the DNA binding domain as has been suggested earlier (Green *et al*, 1995; Zuo *et al*, 1995) and that for this reason we did not detect any Hsc70 binding in this part. Likewise, it might be that upon deletion of 395 to 439, the rest of Hsf1's transactivation domain (440-503) might interact with Gst and thus obscure the binding sites found by us. In general, deletion constructs of proteins without intimate knowledge of their structure could be misleading in interaction studies with proteins like Hsp70s evolved to bind misfolded protein clients.

- Page 14, FigS4A: the sentence 'We simulated this situation and found the corresponding curves do not fit our experimental data' is a bit confusing. I suggest revising as follows: 'However, using this model, the simulated curves did not fit our experimental data'.

We thank the referee for pointing this out and have changed the manuscript accordingly.

- Page 17: the sentence "Nevertheless, Hsc70-mediated.....than for Hsfwt" lack a reference to the corresponding data (Figure 5G).

We thank the reviewer for finding our inadvertent omission and have added this reference in the revised manuscript.

Referee #2:

Summary:

HSF1 activation involves two steps; transition of a monomer to a DNA-binding trimer and acquisition of transcriptional activity. These steps are negatively regulated by chaperones including HSP70 and HSP90, but detailed mechanisms are unclear yet. In this manuscript, Mayer and colleague purified trimeric HSF1 and found using fluorescence polarization assay with Alexa488-labeled DNA that HSF1 trimer bound to DNA was dissociated from it by HSC70. Simultaneously, HSF1 trimer was monomerized. Authors then identified five binding sites of HSC70 within HSF1, which included HR-B proximal site and the site in activation domain, using hydrogen exchange mass spectrometry. Furthermore, three segments encompassing the trimerization domain was deprotected in the presence of HSC70, suggesting that the helices were unfolded. Authors determined that the HR-B proximal site (residues 202-213) was required for HSF1 monomerization by HSC70. Because the trimerization domain contained several potential HSC70 binding sites, which promoted HSF1 monomerization, they proposed that HSC70 monomerizes HSF1 trimer by entropic pulling unzipping of the trimerization domain. Reporter analysis using HSPA6 promoter-driven gene in cells showed that the HR-B proximal site is important for attenuation of the reporter gene transcription. Thus, authors proposed a model of HSC70-mediated monomerization of HSF1 trimer bound to DNA.

Unexpectedly, this study first shows that HSC70 monomerizes HSF1 and dissociates HSF1 from DNA by binding to it directly, and proposes the concept of entropic pulling for HSC70 action on the HSF1 trimerization domain, based on well-controlled and skillful experiments. Their observations greatly extend our understanding of regulation of HSF1 activity and contribute a lot to the field. I have only some concerns to be addressed as below.

We thank the referee for this positive assessment of our study.

Major comments:

1) Authors performed unique in vitro polarization DNA-binding assay, and first showed that HSC70 dissociated HSF1 from DNA (Fig. 1D). Could the same result be observed using Electrophoresis Mobility Shift Assay? Or, they detected HSC70-mediated dissociation because of this unique assay? They should at least comment on it in the text (P7, second paragraph).

We followed this reviewer's suggestion and performed an electrophoretic mobility shift assay yielding similar results. These results are shown in the revised manuscript in extended view Fig EV1C and D. We would like to point out that a fluorescence polarization assay is a homogeneous assay as the reaction is followed in real time, whereas an electrophoretic mobility shift assay is an end point inhomogeneous assay, as after the different incubation times the reaction mixture is loaded onto an agarose gel and then separated by an electric field for around 45 min. The first components of the reaction that are most likely separated from the reaction mixture is ATP and therefore cycling of the Hsp70 machinery will stop and rebinding of released Hsf1 to DNA might occur during analysis. Such a rebinding might be aided by the crowding inside the agarose gel. Notwithstanding, we observed a significant dissociation depending on the incubation time of DNA bound Hsf1 with the Hsp70 machinery.

2) Authors state that "Most of Hsf1 exhibited an electrophoretic mobility that is in between the trimeric and the monomeric state, presumably bound to Hsc70 (P9, line 10)." Are these bands contains HSC70? Authors should examine the presence of HSC70 in these bands, for example, by western blotting using HSC70 antibody or by checking retardation of them after addition of HSC70 antibody.

Although we tried hard to follow this reviewer's suggestions, we were not able to provide unambiguous evidence for our hypothesis that the Hsf1 with an electrophoretic mobility that is in between the trimeric and monomeric state is indeed Hsc70 bound. The reason for this is that in this reaction there were 5 μ M Hsc70 but only 100 nM Hsf1. In addition, Hsc70 is notorious for forming oligomeric species (Wilbanks *et al*, 1995; King *et al*, 1995) and in our blue-native PAGE experiments the Hsc70 band extended upward into the region between the Hsf1 trimer and monomer band even in the absence of Hsf1 and no difference between presence and absence of Hsf1 could be discerned. Likewise, with the antisera available to us we were not able to shift all the Hsc70 sufficiently upward to detect a difference of the 100 nM Hsf1 that might be bound to Hsc70. We therefore weakened our statement and changed the word "presumably" with "possibly" and "bound to Hsc70" to "bound to Hsc70 or DnaJB1 or both" as Hsc70 or DnaJB1 are the only components of the mixture that could bind to and modify the migration of Hsf1 in BN-PAGE.

In addition, did HSF1 dimer appear in the presence of HSC70 as they proposed transient dimerization of HSF1 previously (Hentze et al, 2016)?

This does not seem to be the case. We did not observe a band that is clearly migrating with the same electrophoretic mobility. This is not really surprising, as the dimer band in blue native page usually has an intensity that is less than 10% of the monomer band. With the amount of monomer that we detected in this experiment, we would not be able to detect the dimer.

3) Authors state "To substantiate this hypothesis, we inserted a FLAG epitope between HR-B and the Hsc70 binding site or at 10 and 20 residues distance to HR-B (P16, line 8)." What do this experiment mean? Why Did FLAG antibody disturb HSC70 binding to the HSC70-binding site? Authors should explain it in more detail.

We think there is a misunderstand. In these experiments we tried to dissociate Hsf1 from DNA only by addition of the anti-FLAG antibody halfmer. There was no Hsc70 or DnaJB1 present. The hypothesis was that just binding a mass of about 70 kDa to this site of Hsf1 would be sufficient to dissociate the trimers by entropic pulling. Apparently, this was not the case. One reason was that the trimerization domain is 75 residues long and a single pulling event is not sufficient to unzip the triple-leucin zipper. We have shown in Fig 5B that moving the Hsc70 binding site by 20 residues away from the trimerization domain precludes Hsc70-mediated monomerization of Hsf1. Thus, a single pull may be able to unzip some 10 to 20 residues of the leucin zipper but not more. Then a second Hsc70 will have to bind to continue with the unzipping. To explain this point better we changed the manuscript in the following way:

"To substantiate this hypothesis, we tested whether simple binding of an antibody close to HR-B would be sufficient to unzip the leucine-zipper of Hsf1. We inserted a FLAG epitope between HR-B and the Hsc70 binding site or 10 and 20 residues downstream of HR-B. We treated anti-FLAG antibodies with DTT to split them in half

(Appendix Fig S3) and added them to DNA bound FLAG-epitope containing Hsf1 in the absence of Hsc70 and DnaJB1.”

Of note, although not relevant for this experiment, a FLAG-epitope (DYKDDDDK) would indeed be expected to disturb Hsc70 binding due to the large number of negative charges.

4) Authors state the results shown in Fig. 5 (P17). However, it is difficult to understand it because they did not indicate each HSF1 mutant used in Fig. 5C. For example, reporter activity in the presence of HSF1-202SSS is similar to that in the presence of WT, whereas reporter activity in the presence of HSF1-209SSS is higher. How can we interpret this result?

We are sorry that we omitted in the figure legend to Fig 6C a clear description of the abbreviations we used in the figure for space reasons. In the revised manuscript a clear description has now been added.

When we first saw the cell culture results for the Hsf1-202SSS (Hsf1-I202S,L203S,V205S) we were also very surprised. Even more when realizing that Hsf1 Δ (202-213) exhibited a lower stimulation of heat shock gene transcription than the Hsf1-209SSS (Hsf1-I209S,L211S,L213S) variant. Since the deletion of the Hsc70 binding site had a lower impact than the point mutations, we concluded that the region 202 to 208 must be important for heat shock gene transcription. Indeed, we found that Hsf1 Δ (202-213) needs a higher temperature to transit to the trimeric state than wild type Hsf1 and the Hsf1-209SSS variant. This is now shown in Fig EV3G and H. To solve this issue and to follow this reviewer’s suggestion of point 5 we transfected Hsf1^{-/-}MEFs with plasmids expressing wild type and mutant genes and prepared cell extracts to determine the amount of Hsf1 wild type and mutant proteins by SDS-PAGE and immunoblotting using an Hsf1 specific antiserum. We also determined the amount of trimeric Hsf1 using blue-native PAGE and immunoblotting and we performed *ex vivo* electrophoretic mobility shift assays. For the total amount of Hsf1 wild type and mutant proteins we did not detect any significant differences. However, we detected for the Hsf1 Δ (202-213) and Hsf1-209SSS in each of the three biological repeats more trimers and for Hsf1-209SSS the highest amount of trimer consistent with our finding that this mutant is not monomerized by Hsc70 and with the observation that it had the highest transcriptional activity. Why Hsf1-202SSS and Hsf1 Δ (202-213) do not have such a prominent effect is not clear at the moment and needs further investigations in an independent study, but the reason is not lower protein levels in the transfected cells or inability to trimerize (lower trimerization propensity). In the revised manuscript we discuss this now in more detail.

Which HSF1 mutant represents one lacking HSC70-binding site in the TAD among 4 mutants? Please describe the results more carefully.

We are sorry for not having described this part with greater clarity. We discovered one protected region in the TAD, residues 442-474, that contained two predicted Hsc70 binding sites. The importance of these sites was probed by two Hsf1 variants: Hsf1-L465S,V466S,Y468S (called Hsf1-465SSS in Fig 6) and Hsf1-L473S,F474S,L475S,L476S (called Hsf1-473SSSS). In addition, we tested a previously proposed Hsc70 binding site with two Hsf1 variants (Hsf1 Δ (425-539) and Hsf1-M427S,L429S,L432S,L436S, called Hsf1-427SSSS). In the revised manuscript this is now much more clearly described:

“Analysis of the transfected cells by flow cytometry revealed that deletion or mutation of the HR-B proximal Hsc70 binding site (Hsf1 Δ (202-213) and Hsf1-209SSS) increased the geometric mean of fluorescence of the BFP heat shock reporter by up to threefold in cells with high GFP fluorescence and mutation in the Hsc70 binding

site in the TAD discovered in this study (Hsf1-L465S,V466S,Y468S called Hsf1-465SSS in Fig 6) increased mean BFP fluorescence roughly twofold (Figure 6B, C). Cells with high GFP fluorescence expressing an Hsf1 variant with amino acid replacements in the second site in the TAD with affinity for Hsc70 (Hsf1-L473S,F474S,L475S,L476S called Hsf1-473SSSS) exhibited a 40% increase in BFP fluorescence as compared to cells expressing wild type Hsf1. In contrast, deletion or mutation of the previously proposed Hsc70 binding site in the TAD (Hsf1 Δ (425-439) and Hsf1-M427S,L429S,L432S,L436S called Hsf1-427SSSS in Fig 6C) did not increase mean BFP fluorescence.” In the revised manuscript we performed a much more careful and extensive analysis of the flow cytometry data.

5) In Fig. 5, authors should show oligomeric form of at least some HSF1 mutants in vivo in cells. The results of expression level and in vivo oligomeric state should strength their proposal.

Following this referee’s advice, we determined the average amount of Hsf1 in Hsf1^{-/-} MEFs transiently transfected with plasmids expressing wild type or mutant Hsf1 variants using SDS-PAGE and immunoblotting, determined the oligomeric state of these variants using blue-native PAGE and immunoblotting, and demonstrate DNA binding *ex vivo* using fluorescent-labeled HSE containing DNA oligonucleotides. We also investigated the monomer to oligomer transition temperature *in vitro* for Hsf1 mutants with deletion or mutations adjacent to HR-B. These data are now presented in Fig EV3.

Minor comments:

1) Authors state "These data suggest that binding of Hsc70 to this region contributes to Hsf1 monomerization (P17, line 18)." Please indicate clearly "this region". It is located within the HR-B.

In the revised manuscript we more precisely indicated this region, which is located within HR-B.

2) Fig. 5G should be referred in P17, line 18.

We thank the referee for pointing this out. A reference to Fig 5G has been added in the revised manuscript.

3) In Keywords section, it would be better to include Hsc70. Also, DnaJB1/Hdj1 could be replaced with DnaJB1/Hsp40 or DnaJB1/Hdj1/Hsp40.

Unfortunately, according to the instruction to authors of EMBO Journal the Keywords section only allows up to 5 keywords that should be separated by a slash. Therefore, we excluded DNAJB1/Hdj1/Hsp40 from the Keywords section but included it in the Results section.

4) It would be better to reconsider title to be more specific. For example, HSC70 dissociates HSF1 trimer from DNA by unzipping the trimerization domain.

We thank the referee for this suggestion that we followed.

Referee #3:

Heat shock transcription factor HSF1 is the master regulator of gene expression under a great variety of stress conditions, and recent genome-wide studies on different model systems have revealed its versatile targets in cell stress, growth, development, and disease, thereby extending its physiological impact far beyond the heat shock response or proteotoxic stress response. Yet, many fundamental questions of the activation and de-activation mechanisms of this important transcriptional regulator have remained unanswered in a rigorous manner. In this manuscript, the authors systematically go through past models of the attenuation phase of HSF1 DNA-binding and what evidence the models were based on. The authors then provide with their convincing data on which a new model is based on, while explaining how previous data still fits into their new model. The manuscript presents the minimum components required for HSF1 dissociation from the DNA, and the obtained results will enable a multitude of forthcoming studies to elaborate how these components are regulated in cells to attenuate the heat shock response. In fact, the results in this manuscript reveal that Hsc70-mediated dissociation makes the biggest contribution to attenuation of heat shock gene transcription in a cell culture model. Furthermore, the authors state that it might be the first paper with rigorous tests of the entropic pulling hypothesis mode of Hsp70 action and that their model explains results from many studies from the past 40 years. The impact of this study is certainly high, but the authors could make it easier for the reader to understand how important the results are. Now, the importance is mentioned quite briefly in the Discussion only.

We thank this referee for the very positive assessment of our work.

The manuscript is mostly well written, but the "modest" Abstract and the end of Introduction do not make justice for the content and impact of the study.

Unfortunately, the EMBO Journal style only allows 175 words for the abstract. Since our abstract has already 174 words, more emphasis on the importance of our findings could not be added to this part. Instead, following the referee's advice we added some emphasis of the importance of our work to the end of the introduction.

Since the methodology of the study is the key for mechanistic understanding of the HSF1 attenuation mechanism, the authors could explain the basis of the fluorescence polarization assay in a more detail.

Following the referee's suggestion, we added this explanation to the revised manuscript: "Fluorophores excited by polarized light, emit light with the same polarization plane. If the fluorophore rotates between excitation and emission, the plane of polarization rotates with the dye. Therefore, fluorescence polarization monitors the rotational diffusion of a fluorescent molecule, attached to DNA in our case. Rates of rotational diffusion are high and therefore fluorescence polarization is low for free DNA. Binding of Hsf1 to the DNA decreases the rate of rotational diffusion and consequently increases fluorescence polarization."

Specific comments:

1. The figure legend of Figure 2 is hard to follow, especially for panel C.

We tried to improve the figure legend to make this complicated experiment more accessible.

2. In Figure 4A, it is possible to distinguish five or six peaks with value below -5. On page 11, it is stated that "...two of the segments (200-213 and 442-474) protected from hydrogen exchange by Hsc70 covered sequences that fitted properties of strong Hsp70 binding sites...", which is clear but why are the other peaks at -5 not addressed?

It would not be surprising, if all the segments with a score <-5 would turn out to be Hsc70 binding sites. In fact, most of them will be binding sites. However, not all segments with a prediction score of <-5 are necessarily Hsc70 binding sites in the **folded protein**. They might be bound by Hsc70, if Hsf1 is unfolded or during *de novo* synthesis of Hsf1 at the ribosome. All sites for which we did not observe any protection from hydrogen exchange in the presence of Hsc70 we considered unlikely to be accessible to Hsc70 binding in the folded state of Hsf1. In fact, a high density of predicted Hsc70 binding sites is found in the trimerization region, which is not surprising since the trimerization region consists of a leucine zipper and leucine is the preferred residue for Hsc70 binding. In addition, the trimerization domain is rich in positive charges, also favoring Hsc70 binding. In general, the hydrophobic core of a protein contains predicted Hsc70 binding sites. For this reason, Hsp70s bind promiscuously to most proteins when they are unfolded or during *de novo* synthesis on the ribosome or during translocation through membranes.

We addressed the 5 regions that exhibited an appreciable protection in HX-MS experiments shown in Fig 3, 78-98, 200-213, 229-252, 347-375, 442-474. For each region we used the Hsp70 binding algorithm to detect 13-mer segments with a score below -5. Two of the segments (202-213 and 442-474) contained two potential Hsc70 binding sites each. The other segments did not contain predicted Hsc70 binding sites. Not all sites protected in HX experiments in the presence of Hsc70 are necessarily Hsc70 binding sites, as Hsc70 could induce allosterically conformational changes that could lead to protection and deprotection as has been observed in several examples (Rodriguez *et al*, 2008; Boysen *et al*, 2019). This is now explained in more detail in the revised manuscript.

3. In Figure 4B, four peptides display $>>60 \mu\text{M}$. Does this refer to the peptides described in the text "...To the other protected segments Hsc70 did not have any measurable affinity...?"

This is partially correct. However, we were a bit too condensed in this statement.

If this is the case then why does the text refer to "For all four potential binding sites we could determine a KD between ...", when the figure displays five peptides that have a measurable Kd. Please, provide a more clear description of these results.

We are sorry that we did not explain this experiment in sufficient detail. In the revised manuscript we have now amended the figure legend: "Peptides N74-Q86-C (NMYGFRKVVHIEQC), P223-H235-C (PKYSRQFSLEHVHC) and A343-R352-C (ASVTALTDARC) represent the most hydrophobic part of HX protected segments 78-98, 229-252, and 347-375, respectively; N200-K208-C (NRILGVKRKC), N200-K208-C_SSS (NRSSGSKRKC), G204-N214-C (GVKRKIPLMLNC), and G204-N214-C_SSS (GVKRKSPMSNC) represent the two potential Hsc70 binding sites in segment 202-213 in original sequence or with hydrophobic residues replaced by serine; S461-Q471-C (SGKQLVHYTAQC) and Y468-S480-C (YTAQPLFLLDPGSC)

represent the two potential Hsc70 binding sites in segment 442-474” and the main text: “For all four potential binding sites in the two HX protected regions 202-213 and 442-474 we could determine a K_D between 5 and 30 μM (Figure 4B). To peptides representing the most hydrophobic part of the other protected segments (N74-Q86-C, P223-H235-C, A343-R352-C) Hsc70 had a lower or no measurable affinity.” It should be mentioned that affinities of Hsc70 for a peptide with K_D 's above 40 μM become more and more unreliable as Hsc70 has a strong tendency to oligomerize at high concentrations.

4. In the Results section "The number of available Hsc70 binding sites determines the rate of Hsf1 monomerization" (p. 13-14), the description of experimental set-ups and results is difficult to understand. This part of the text needs improving considering the importance of the obtained results.

We tried to make this part clearer by adding some more explanations in the revised manuscript.

5. Although the experiments are mainly performed *in vitro*, there is one critical experiment linking the results to a cell-based model system. As shown in Figure 6, HSF1^{-/-} MEFs were stably transfected with a heat shock reporter and subsequently they were transiently transfected to express either wild-type HSF1 or HSF1 where the HR-B (within the trimerization domain of HSF1) proximal Hsc70-binding site or the TAD (transactivation domain of HSF1) distal Hsc70-binding site was disrupted. Based on the results from the reporter assay, the authors conclude that binding of Hsc70 to the HR-B proximal Hsc70-binding site is important for attenuation of heat shock gene transcription, whereas they suggest that the Hsc70-binding site in the TAD contributes to attenuation indirectly, presumably through interference of HSF1 interaction with the core transcription machinery. This conclusion is intriguing and raises an important question why the experiment shown here was performed in unstressed cells only. It would be important to perform a corresponding experiment in cells exposed to heat shock and followed by a recovery. This experiment would certainly provide more rigor to the proposed model of attenuation.

We had originally performed a heat shock experiment as suggested by the referee but did not report them in the original manuscript because we thought they would not add to our conclusions and we did not quite understand the results and the reason why we did not observe a heat shock induced increase in BFP fluorescence. In response to this referee's comments we performed additional experiments and a much more rigorous analysis of the flow cytometry data.

But first we would like to explain why we used a flow cytometry assay for this analysis and not the generally used luciferase reporter assay. MEFs in general and Hsf1^{-/-} MEFs in particular are difficult to transfect and transfection efficiency is low. For two reasons we did not want to use a luciferase reporter assay. First, in transient transfection it cannot be guaranteed that all cells receive similar amounts of reporter and Hsf1 expressing plasmid DNA which could obscure what is actually going on in the cells and averages out difference between cells. Second, firefly luciferase commonly used in reporter assays is one of the model substrates for Hsp70 (Schröder *et al*, 1993) and denatures and becomes inactive under heat shock but is refolded by Hsp70 upon return to 37°C also *in vivo* (Nollen *et al*, 1999). Therefore, the luciferase activity measured during the recovery phase reports on newly synthesized luciferase due to heat shock transcription but also on refolding of pre-existing luciferase denatured during heat shock. BFP like GFP is a stable protein that has a thermal

unfolding transition temperature of 85°C (Topell *et al.*, 1999) and thus does not unfold during heat shock. As second advantage of a flow cytometry system we considered that we would be able to distinguish transfected from non-transfected cells. What we realized during analysis of the data is that the GFP positive cells do not form a population of cells that is clearly distinguishable from the non-transfected cell population. In the contrary, GFP fluorescence distributes over 3 orders of magnitude from the autofluorescence indistinguishable of non-transfected cells (around 100 units in MEFs) to 10^5 units. According to the manufacturer of the flow cytometry instrument the fluorescence is linear in a range between around 40 and $2.4 \cdot 10^5$ (regularly verified by fluorescent beads) and the amount of GFP scales linearly with fluorescence. Since we expressed Hsf1 and GFP from a polycistronic mRNA using an internal ribosomal entry site (IRES) we have to conclude that Hsf1 also varies by up to three orders of magnitude in abundance and that this is an intrinsic feature of transient transfection using lipofectamine reagents or electroporation (one of the replicates was performed by electroporation with similar results); we also observed this for two different vector controls and for transfected HeLa cells (see below). Furthermore, since Hsf1 trimerization does not only depend on temperature but also on Hsf1 concentration, as we found earlier (Hentze *et al.*, 2016), it is not surprising that the cells that exhibit high GFP fluorescence and therefore have high Hsf1 concentrations do not respond to heat shock anymore. They have high levels of trimeric Hsf1 that increase the cellular levels of Hsp70 and high levels of Hsp70 was shown earlier to prevent the induction of the heat shock response (Baler *et al.*, 1996). Consistently, we observed that cells transfected with Hsf1wt exhibited a linear correlation of BFP and GFP fluorescence up to an upper level of BFP fluorescence. We therefore examined the cells in the GFP_{low} gate, which included cells with 40 to 10^3 units of GFP fluorescence and also contained the untransfected cells (Fig 6E, F). It should be noted that in the previous version of the manuscript we reported the median of fluorescence. For values distributed over several orders of magnitude the arithmetic mean is not a good measure since single extremely high values skews the results and the median is a more robust measure for the average of the population since 50% of the values are below and 50% above the median. This is completely adequate for the cells in the GFP_{high} gate. However, for the GFP_{low} gate the median is completely dominated by the untransfected cells as the transfection efficiency is below 50%. We therefore report now the geometric mean

$y = (x_1 \cdot x_2 \cdot \dots \cdot x_n)^{\frac{1}{n}}$, which is also robust against extreme values but is not dominated as much by the non-transfected cells. Although for the GFP_{low} gate a significant difference between Hsf1wt and Hsf1-209SSS was observed, heat shock did not further increase BFP fluorescence. To exclude the non-transfected cells, we analyzed cells in a subgate of the GFP_{low} gate with a BFP fluorescence higher than 10^4 (GFP_{low}/BFP_{high}). These values revealed the real effect of the mutation in the Hsc70 binding site but again we did not observe heat shock induction even in the presence of wild type Hsf1. Obviously, even low concentrations of Hsf1 did not increase heat shock transcription in response to heat stress. The reason might be that transfection with lipofectamine or electroporation already stressed the MEFs to an extent that this leads to BFP and of course also Hsp70 expression such that a second stress does not increase transcription further. Also, BFP is not degraded and the MEFs do not divide sufficiently to decrease the initial BFP and Hsp70 levels.

To analyze our mutants in a more robust cell line we stably transfected HeLa cells with our BFP reporter and treated and analyzed these cells in a similar way as the MEFs. Since HeLa cells have endogenous Hsf1 cells transfected with the vector control exhibited a heat shock induction. We realized that the average BFP fluorescence of the cells transfected with Hsf1 encoding plasmids in the GFP_{high} gate under non-stress conditions is already higher than the BFP fluorescence in the vector control after heat shock, indicating that the Hsf1 concentration in these cells were already high enough to trimerize and drive heat shock gene

transcription. Consequently, the Hsp70 concentrations in the cells must also be higher and it is not surprising that we did not observe a heat shock induction in this population. However, in the GFP_{low} cells we observed heat shock induction for all Hsf1 variants with significant increase for variants defective in Hsc70 binding.

It should also be noted that the equilibrium of high levels of Hsf1 trimerization caused by overexpression of Hsf1 and consequently high expression of Hsp70 and repression of transcriptional activity of Hsf1 by Hsp70 is conceptually similar to the situation during the attenuation phase during a heat shock when hsf1 continuously trimerizes due to its thermosensory function and is continuously monomerized by high levels of expressed Hsp70. All of this is now carefully described and discussed in the revised manuscript.

6. In the end of the Discussion, it is postulated that "In particular, phosphorylation which was shown to accompany heat shock induction and to prolong heat shock gene transcription is expected to interfere with Hsc70 binding due to the fact that negative charges disfavor Hsp70 binding (Rudiger et al. 1997)". However, in the study by Budzyński et al. 2015 (doi: 10.1128/MCB.00816-14), it was shown that phosphorylation within the so-called regulatory domain (RD) is not required for HSF1 activation. It would be interesting to know how the authors think of a possibility that a lack of phosphorylation is able to promote or facilitate HSF1 activation, keeping in mind that less phosphorylation should in this case attract more Hsc70 binding.

This is indeed a very interesting issue and the influence of PTMs not only phosphorylation on the dissociation and Hsf1 activity should be addressed using our system in the near future. Phosphorylation/PTM in close vicinity of the Hsp70 binding site could influence Hsp70 binding assuming that the modification does not change protein conformation. In the mentioned study Budzyński et al. the mutated phosphorylation sites were in the regulatory region between 220 to 389, an unstructured region that is unlikely to change its conformation upon modification. This region is outside the sites where Hsc70 bound in our study to monomerize Hsf1 (202-213) or to attenuate interaction with the transcriptional machinery (465-473). The findings of this study therefore is not in conflict with our hypothesis.

References

- Baler R, Zou J & Voellmy R (1996) Evidence for a role of Hsp70 in the regulation of the heat shock response in mammalian cells. *Cell Stress & Chaperones* **1**: 33–39
- Boysen M, Kityk R & Mayer MP (2019) Hsp70- and Hsp90-Mediated Regulation of the Conformation of p53 DNA Binding Domain and p53 Cancer Variants. *Molecular Cell* **74**: 831–843.e4
- Green M, Schuetz TJ, Sullivan EK & Kingston RE (1995) A heat shock-responsive domain of human HSF1 that regulates transcription activation domain function. *Molecular and Cellular Biology* **15**: 3354–3362
- Hentze N, Le Breton L, Wiesner J, Kempf G & Mayer MP (2016) Molecular mechanism of thermosensory function of human heat shock transcription factor Hsf1. *Elife* **5**: e11576
- Jiang Y, Rossi P & Kalodimos CG (2019) Structural basis for client recognition and activity of Hsp40 chaperones. *Science* **365**: 1313–1319

- King C, Eisenberg E & Greene L (1995) Polymerization of 70-kDa heat shock protein by yeast DnaJ in ATP. *J Biol Chem* **270**: 22535–22540
- Morshauer RC, Hu W, Wang H, Pang Y, Flynn GC & Zuiderweg ER (1999) High-resolution solution structure of the 18 kDa substrate-binding domain of the mammalian chaperone protein Hsc70. *Journal of Molecular Biology* **289**: 1387–1403
- Nollen EA, Brunsting JF, Roelofsen H, Weber LA & Kampinga HH (1999) In vivo chaperone activity of heat shock protein 70 and thermotolerance. *Molecular and Cellular Biology* **19**: 2069–2079
- Rodriguez F, Arsène-Ploetze F, Rist W, Rüdiger S, Schneider-Mergener J, Mayer MP & Bukau B (2008) Molecular basis for regulation of the heat shock transcription factor sigma32 by the DnaK and DnaJ chaperones. *Molecular Cell* **32**: 347–358
- Rüdiger S, Schneider-Mergener J & Bukau B (2001) Its substrate specificity characterizes the DnaJ co-chaperone as a scanning factor for the DnaK chaperone. *EMBO J* **20**: 1042–1050
- Schröder H, Langer T, Hartl FU & Bukau B (1993) DnaK, DnaJ and GrpE form a cellular chaperone machinery capable of repairing heat-induced protein damage. *EMBO J* **12**: 4137–4144
- Topell S, Hennecke J & Glockshuber R (1999) Circularly permuted variants of the green fluorescent protein. *FEBS LETTERS* **457**: 283–289
- Wilbanks SM, Chen L, Tsuruta H, Hodgson KO & McKay DB (1995) Solution small-angle X-ray scattering study of the molecular chaperone Hsc70 and its subfragments. *Biochemistry* **34**: 12095–12106
- Zhu X, Zhao X, Burkholder WF, Gragerov A, Ogata CM, Gottesman ME & Hendrickson WA (1996) Structural analysis of substrate binding by the molecular chaperone DnaK. *Science* **272**: 1606–1614
- Zuo J, Rungger D & Voellmy R (1995) Multiple layers of regulation of human heat shock transcription factor 1. *Molecular and Cellular Biology* **15**: 4319–4330

Prof. Matthias P. Mayer
University of Heidelberg
ZMBH
Im Neuenheimer Feld 282
Heidelberg, Baden-Wuerttemberg 69120
Germany

9th Apr 2020

Re: EMBOJ-2019-104096R
Hsp70s dissociate Hsf1 trimers from DNA by unzipping the trimerization domain

Dear Prof. Mayer,

Thank you for submitting your revised manuscript for our consideration. Please apologize the delay in communicating this decision to you, which was due to delayed referee reports on account of the current Covid-19 pandemic. We now have the reports from two of the original referees (see comments below). I am pleased to say that the referees find that their comments have been satisfactorily addressed. Referee #1 suggests some textual changes, which can be added in a final revised version. I would now also ask you to address a number of editorial issues that are listed in detail below in this final version. Please make any changes to the manuscript text in the attached document only using the "track changes" option. Once these remaining issues are resolved, we will be happy to formally accept the manuscript for publication.

Thank you again for giving us the chance to consider your manuscript for The EMBO Journal. I look forward to receiving your final revision. Please feel free to contact me if you have further questions regarding the revision or any of the specific points listed below.

Kind regards,

Stefanie Boehm

Stefanie Boehm
Editor
The EMBO Journal

Referee #1:

The authors have (more than) adequately addressed all the points I raised in my initial report. As final remark, I would like to stress that the revised text on the cellular experiments (with HSF1^{-/-} and HeLa cells) is rather complex and fuzzy written. I would suggest the authors to simplify this a bit for clarity.

Referee #2:

The authors performed a lot of works to respond to my concerns, and carefully and intensively revised the manuscript. I am sure that their observations greatly extend our understanding of regulation of HSF1 activity and contribute a lot to the field.

Referee #1:

The authors have (more than) adequately addressed all the points I raised in my initial report.

We thank this referee for the positive assessment of our revision.

As final remark, I would like to stress that the revised text on the cellular experiments (with HSF1-/- and HeLa cells) is rather complex and fuzzy written. I would suggest the authors to simplify this a bit for clarity.

As suggested by this referee we rewrote the flow cytometry part of the result section and hope that now it is easier accessible.

Referee #2:

The authors performed a lot of works to respond to my concerns, and carefully and intensively revised the manuscript. I am sure that their observations greatly extend our understanding of regulation of HSF1 activity and contribute a lot to the field.

We thank this referee for the positive assessment.

Prof. Matthias P. Mayer
University of Heidelberg
ZMBH
Im Neuenheimer Feld 282
Heidelberg, Baden-Wuerttemberg 69120
Germany

24th Apr 2020

Re: EMBOJ-2019-104096R1

Feedback regulation of heat shock factor 1 (Hsf1) activity by Hsp70-mediated trimer unzipping and dissociation from DNA

Dear Matthias,

Thank you again for submitting the final revised version of your manuscript. I am pleased to inform you that we have now accepted it for publication in The EMBO Journal.

Your article will be processed for publication in The EMBO Journal by EMBO Press and Wiley, who will contact you with further information regarding production/publication procedures and license requirements.

Should you be planning a Press Release on your article, please get in contact with embojournal@wiley.com as early as possible, in order to coordinate publication and release dates.

Please also be aware that under the DEAL agreement of German scientific institutions with our publisher Wiley, you could be eligible for free publication of your article in the open access format (<https://authorservices.wiley.com/author-resources/Journal-Authors/open-access/affiliation-policies-payments/german-projekt-deal-agreement.html>). Please contact either the administration at your institution or our publishers at Wiley (embojournal@wiley.com) for further questions.'

Congratulations on your successful publication, and thank you again for this contribution to The EMBO Journal! Please continue to consider EMBO Journal for your work in the future.

Kind regards,

Stefanie

Stefanie Boehm
Editor
The EMBO Journal

Corresponding Author Name: Matthias P. Mayer

Manuscript Number: EMBOJ-2019-104096